# Dynamic mechanisms of CRISPR interference by *Escherichia coli* CRISPR-Cas3

Kazuto Yoshimi [1,2,7], Kohei Takeshita[3,7], Noriyuki Kodera [4,7], Satomi Shibumura[5], Yuko Yamauchi[1], Mine Omatsu[3,6], Kenichi Umeda[4], Yayoi Kunihiro[5], Masaki Yamamoto[3,6] & Tomoji Mashimo [1,2] ✉

Type I CRISPR-Cas3 uses an RNA-guided multi Cas-protein complex, Cascade, which detects and degrades foreign nucleic acids via the helicase-nuclease Cas3 protein. Despite many studies using cryoEM and smFRET, the precise mechanism of Cas3-mediated cleavage and degradation of target DNA remains elusive. Here we reconstitute the CRISPR-Cas3 system in vitro to show how the *Escherichia coli* Cas3 (EcoCas3) with EcoCascade exhibits collateral non-specific single-stranded DNA (ssDNA) cleavage and target specific DNA degradation. Partial binding of EcoCascade to target DNA with tolerated mis-matches within the spacer sequence, but not the PAM, elicits collateral ssDNA cleavage activity of recruited EcoCas3. Conversely, stable binding with complete R-loop formation drives EcoCas3 to nick the non-target strand (NTS) in the bound DNA. Helicase-dependent unwinding then combines with *trans* ssDNA cleavage of the target strand and repetitive *cis* cleavage of the NTS to degrade the target double-stranded DNA (dsDNA) substrate. High-speed atomic force microscopy demonstrates that EcoCas3 bound to EcoCascade repeatedly reels and releases the target DNA, followed by target fragmentation. Together, these results provide a revised model for collateral ssDNA cleavage and target dsDNA degradation by CRISPR-Cas3, furthering understanding of type I CRISPR priming and interference and informing future genome editing tools.

The clustered-regularly-interspaced-short-palindromic-repeats (CRISPR) CRISPR-associated proteins (Cas) system allows for adaptive immunity in prokaryotes. CRISPR protein complexes comprise two classes, with each class classified into three types where Class 1 includes types I, III, IV and Class 2 includes types II, V, and VI[1]. Class 1 systems use multiple different Cas proteins, while Class 2 effectors contain only a single protein. To date, much attention has focused on the mechanism of Class 2 effectors, such as type II Cas9, type V Cas12, and type VI Cas13, given

their practical applications in genome editing and manipulation[2–6]. Type 1 systems are also now emerging as tools for genome and transcriptome manipulation in microbiota[7,8] and eukaryotic cells[9–11]. Binding of type I CRISPR-Cas effectors to DNA sequences in the absence of Cas3 leads to transcriptional repression in bacteria[7] and human cells[9]. For DNA editing, introduction of the Cascade multi Cas-protein complex, CRISPR RNA (crRNA), and the Cas3 helicase-nuclease into mammalian cells results in long-range chromosomal deletions in target DNA[10,11]. The

[1]Division of Animal Genetics, Laboratory Animal Research Center, Institute of Medical Science, University of Tokyo, Tokyo 108-8639, Japan. [2]Division of Genome Engineering, Center for Experimental Medicine and Systems Biology, Institute of Medical Science, University of Tokyo, Tokyo 108-8639, Japan. [3]Life Science Research Infrastructure Group, Advanced Photon Technology Division, RIKEN SPring-8 Center, Hyogo 679-5148, Japan. [4]Nano Life Science Institute (WPI-NanoLSI), Kanazawa University, Kakuma-machi, Kanazawa 920-1192, Japan. [5]C4U Corporation, Osaka 565-0871, Japan. [6]Laboratory of Macromolecular Dynamics and X-ray Crystallography, Department of Life Science, University of Hyogo, Hyogo 678-1297, Japan. [7]These authors contributed equally: Kazuto Yoshimi, Kohei Takeshita, Noriyuki Kodera. ✉e-mail: mashimo@ims.u-tokyo.ac.jp

long-range deletions generated by Cas3 contrasts with smaller deletions (or indels) generated by Cas9/Cas12 editing, and has led to the descriptions of DNA shredder and scissors, respectively.

Considerable efforts have been devoted to understanding the mechanism of CRISPR interference by type I CRISPR[12–27]. Several cryo-electron microscopy (EM) structures of type I CRISPR complexes have been solved, revealing seahorse-shaped structures containing Cas5, Cas6, multiple Cas7, Cas8 (Cse1), which recognizes the PAM, and two Cas11 (Cse2) (Fig. 1a). Type I CRISPR systems target homologous regions of double-stranded DNA (dsDNA) for degradation through two major steps: recognition of a target DNA by Cascade complex surveillance, and cleavage of the DNA by Cas3 that is recruited by the Cascade complex[24–26]. In the first step, the Cascade complex, including Cas8, scans the PAM (protospacer adjacent motif) and initiates DNA

unwinding at the PAM[28]. Subsequently, crRNA hybridization with the target DNA strand (TS) leads to displacement of the non-target strand (NTS), forming a three-stranded nucleic acid structure known as an R-loop[14–16,18,19,27]. Complete formation of the R-loop induces a conformational change in the Cascade complex that enables recruitment of Cas3[15,19,29]. The recruited Cas3, a protein with an SF2 (Superfamily 2) helicase domain and a HD (histidine-aspartate) nuclease domain, degrades the target DNA in a unidirectional ATP-dependent manner according to the following steps: nicking the NTS at the R-loop, loading onto the ssDNA, and unwinding the DNA while degrading the DNA[13,15,26,30,31]. Recently, single-molecule Förster resonance energy transfer (smFRET) experiments have shown that Cas3 can remain associated with Cascade to cleave ssDNA by a reeling mechanism[12]. However, Cas3 can also break free of Cascade and translocate on its

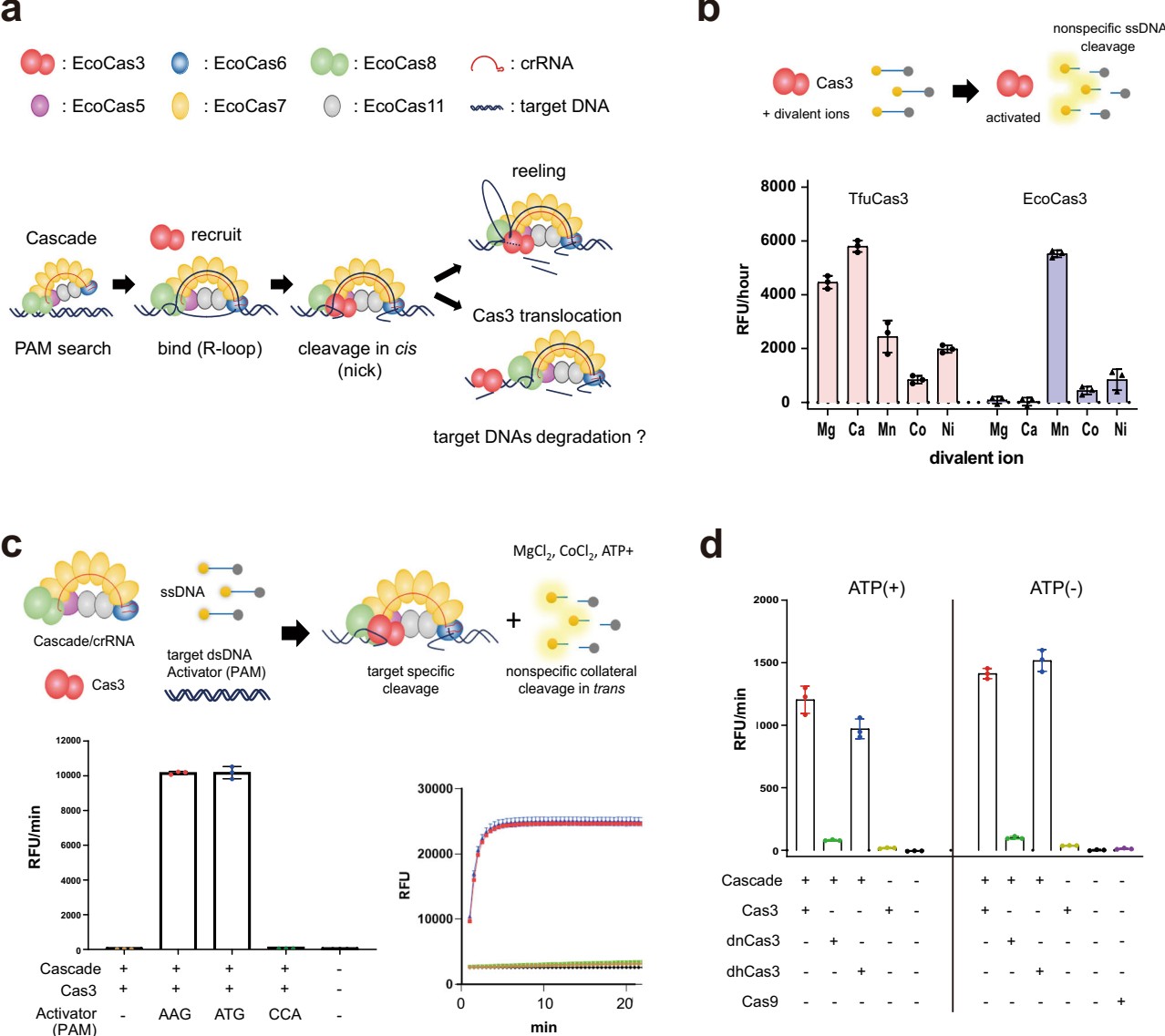

**Fig. 1 | In vitro reconstituted EcoCas3-EcoCascade/crRNA complex cleaves nonspecific ssDNA in trans. a** Schematic depiction of the known type I CRISPR interference mechanism. **b, c** Electrophoretic mobility shift assay (EMSA) and DNA degradation assay. **b** Activation of EcoCas3 and TfuCas3 by divalent metal ions ($Mg^{2+}$, $Ca^{2+}$, $Mn^{2+}$, $Co^{2+}$, and $Ni^{2+}$). Fluorescent dye-quencher (FQ)-labeled ssDNA probes measured promiscuous ssDNA cleavage activity. RFU: relative fluorescence unit. **c** Collateral ssDNA cleavage activity measured by incubation of EcoCas3-EcoCascade/crRNA complex with a 60 bp dsDNA Activator containing a target sequence flanked by a PAM and an FQ-labeled ssDNA probe in reaction buffer

containing $MgCl_2$, $CoCl_2$, and ATP for 10 min at 37 °C. Quantitatively represented by RFU per min (left) or RFU at 10 min (right). **d** EcoCas3 HD domain H74A (dead nuclease mutant, dnCas3), abolished collateral cleavage activity, while SF2 motif III S483A/T485A (dead helicase mutant, dhCas3) showed collateral cleavage activity. Collateral activity in ATP reaction buffer (+) was at the same level as that in ATP-free buffer (−) for wild-type EcoCas3 and the dhCas3 mutant. SpCas9 did not exhibit any collateral cleavage activity. Data in **b**–**d** are presented for $n = 3$ independent measurements and mean value, error bars represent SD values. Source data are provided as a Source Data file.

own through the target DNA[20]. It is unclear whether Cas3 degrades DNA during independent translocation and how Cas3 with a single HD domain can degrade both the NTS and the TS of dsDNA[32,33] (Fig. 1a).

This study reveals that the *Escherichia coli* Cascade crRNA complex (EcoCascade) assembled with *E. coli* Cas3 (EcoCas3) exhibits collateral *trans*-cleavage activity on a non-specific ssDNA. This promiscuous cleavage of collateral ssDNAs, named *trans*-cleavage, is distinguished from the programmable on-target cleavage of target ssDNA, namely, *cis*-cleavage[34–36] (Fig. 1a). We show that unstable Eco-Cascade binding with partial R-loop formation with EcoCas3 mediates the collateral *trans*-cleavage, but does not lead to double-stranded DNA cleavage. In contrast, stable EcoCascade binding with locked R-loop construction provides *cis*-cleavage of the NTS with helicase-dependent *trans* separation and cleavage of the TS, resulting in progressive degradation of target dsDNA substrates. In vitro experiments using high-speed atomic force microscopy (hs-AFM) also demonstrate that EcoCas3 remains tightly associated with the EcoCascade, which repeatedly reels and releases the target DNA, followed by target degradation. These findings provide insight into the mechanism of type I CRISPR-Cas3 priming and interference against a foreign DNA.

## Results

### In vitro reconstitution of *Escherichia coli* CRISPR-Cas3 interference

*E. coli* CRISPR-Cas3 is generally well-characterized type I CRISPR complexes in vitro and in vivo[32,33,37,38]. However, recombinant EcoCas3 protein is difficult to purify because of poor solubility and propensity to aggregate at 37 °C[25,26,30,39]. Co-expression of HtpG chaperon[40] and/or low temperature growth at 20 °C[24,25] produced only a limited amount of protein that was highly aggregated (Supplementary Fig. 1a). This is in contrast to isolated *Thermobifida fusca* Cas3 (TfuCas3) protein produced by the *E. coli* bacterial expression system at 37 °C[10,15,31] (Supplementary Fig. 1b). We then used Sf9 insect cells with a baculovirus expression system at 20 °C to produce EcoCas3 protein, which was soluble and ~95% homogeneous (Supplementary Fig. 2a, b). Eco-Cascade proteins and crRNA were co-expressed in *E. coli* JM109(DE3) and purified using Ni-NTA resin as previously reported[24,41] (Supplementary Fig. 2c, d). Tycho NT.6 protein stability measurements (Supplementary Fig. 3) and the ProteoStat protein aggregation assay (Supplementary Fig. 4) indicated that the temperature-dependent stability and aggregation onset temperature of EcoCas3 was consistent with a mesophilic protein[25,26,30,39]. Then, we confirmed co-purified recombinant EcoCascade-crRNA ribonucleoproteins (RNPs) bound to supercoiled (SC) plasmids composed of *hEMX1* spacer sequences flanked by a PAM (5′-AAG-3′), but not bound to SC plasmids including spacer sequences flanked by a nonPAM (5′-CCA-3′) (Supplementary Fig. 5a). Binding of EcoCascade-crRNA RNPs was also observed for linear dsDNA molecules (*mTyr* and *rIl2rg* genes) (Supplementary Fig. 5b), and exchanging nucleotide pairs between crRNAs and target sequences abolished this binding (Supplementary Fig. 5c). Finally, assembly of EcoCascade RNPs with purified EcoCas3 protein specifically degraded SC plasmids (*hEMX1*) and linear dsDNA (*mTyr*) in the presence of ATP and Mg²⁺ [24,25,30,39,42] (Supplementary Fig. 6a, b).

### The EcoCas3-EcoCascade-crRNA complex nonspecifically cleaves ssDNAs in *trans*

Cas3 proteins from *Streptococcus thermophiles, Methanocaldococcus jannaschii*, and *Thermus thermophilus* can exhibit indiscriminate, divalent cation-dependent ssDNase activity in the absence of Cascade[30,39,42]. Using fluorescent dye-quencher (FQ)-labeled ssDNA probes, we found that EcoCas3 and TfuCas3 also exhibit nonspecific ssDNA cleavage in a metal-dependent manner, although the dependency was different between the two bacteria (Fig. 1b). TfuCas3 cleaved ssDNA with all divalent ions tested, whereas EcoCas3 was only activated with Mn²⁺ and Ni²⁺, consistent with previous results[25].

Type V Cas12a, an RNA-guided DNase[34], and type VI Cas13, an RNA-guided RNase[43], engage in collateral cleavage of nearby non-specific nucleic acids after their targeted activity. To investigate whether Cas3 also possesses collateral ssDNA cleavage activity, we assembled EcoCas3, EcoCascade RNPs, 60 bp dsDNA fragments containing target sequences flanked by a PAM (targeted Activator), and a untargeted ssDNA[44]. We found that targeted degradation triggered untargeted degradation of both circular M13 phage ssDNA and linearized long ssDNA, but not of circular pBlueScript dsDNA (Supplementary Fig. 7a). As is the case with target dsDNA degradation by EcoCascade RNPs and EcoCas3 (Supplementary Fig. 6a, b), this collateral ssDNA cleavage was dependent upon the presence of a PAM in the targeted nucleic acid (Supplementary Fig. 7a). These results indicate that either some metal ions or Cascade target-binding by R-loop formation can induce EcoCas3-dependent non-specific ssDNA cleavage activity in vitro.

To quantitatively measure collateral ssDNA cleavage activity we used a FQ-labeled untargeted ssDNA probe[34,43] (Fig. 1c), which is used in CRISPR-based diagnostics as a platform for rapid and sensitive nucleic acid detection, for example in Covid-19 test kits[45–47]. Consistent with the results of the M13/linear ssDNA cleavage (Supplementary Fig. 7a), EcoCas3 showed collateral ssDNA cleavage in a PAM-dependent manner (with a PAM of 5′-AAG-3′ or 5′-ATG-3′, but not 5′-CCA-3′) (Fig. 1c and Supplementary Fig. 7b, c). Fluorescent reporter DNA oligonucleotides (DNaseAlert™ IDT) also confirmed this collateral cleavage activity (Supplementary Fig. 8a), whereas fluorescent reporter RNA oligonucleotides (RNaseAlert™, IDT) detected little or no collateral RNase activity (Supplementary Fig. 8b). We previously showed that mutants of EcoCas3 in the HD domain (H74A, dead nuclease, dnCas3), and SF2 motif III (S483A/T485A, dead helicase, dhCas3) abolished target DNA degradation in human cells[11]. In the collateral cleavage assay, the dnCas3 mutant abolished all cleavage activity, but the dhCas3 mutant showed a similar level of activity as that of wild-type EcoCas3 (Fig. 1d). In ATP-free reaction buffer (−), the collateral activity of the EcoCas3 protein was at the same level or higher compared with that of wild-type EcoCas3 and the dhCas3 mutant in ATP (+) buffer (Fig. 1d). Together, these results indicate that *trans* cleavage of non-specific ssDNA is catalyzed by the single HD domain of EcoCas3 and that the helicase activity is not required for the *trans* cleavage but may restrain it.

### PAM recognition is a prerequisite for collateral ssDNA cleavage by Cas3 but not Cas12a

Having determined that collateral ssDNA cleavage by the EcoCas3-EcoCascade complex is PAM-dependent (Fig. 1c), we sought to further characterize the specificity of PAM recognition by screening all 64 possible target sites containing each of the three-nucleotide PAM sequences (Fig. 2a and Supplementary Fig. 9a). We observed collateral cleavage activity with 14 PAM types, with the highest activity from 5′-AAG-3′ and 5′-ATG-3′, followed by 5′-GAG-3′, 5′-AAA-3′, 5′-AAC-3′, 5′-TAG-3′, and 5′-AGG-3′. There was no cleavage when the first or second PAM nucleotide was C or the third nucleotide was T (Fig. 2a and Supplementary Fig. 9a). This PAM recognition specificity for *trans* cleavage activity matched the results from an in vivo high-throughput CRISPR-interference assay[48]. In contrast, LbaCas12a showed collateral cleavage activity with almost all 64 PAM types, with the highest activity with 5′-GGGG-3′ and the lowest with 5′-GCCG-3′ (Fig. 2b and Supplementary Fig. 9b).

According to previous reports[34–36], binding of the ssDNA complementary to the crRNA activates Cas12a for nonspecific *trans* cleavage. We also observed that EcoCas3 and LbaCas12a were activated by crRNA-complementary ssDNA (TS) but not by non-complementary ssDNA (NTS) (Fig. 2c, d). However, the PAM specificity was different between EcoCas3 and LbaCas12a. LbaCas12a was activated by both crRNA-complementary TS flanked by a PAM (3′-AAAC-5′) or a nonPAM

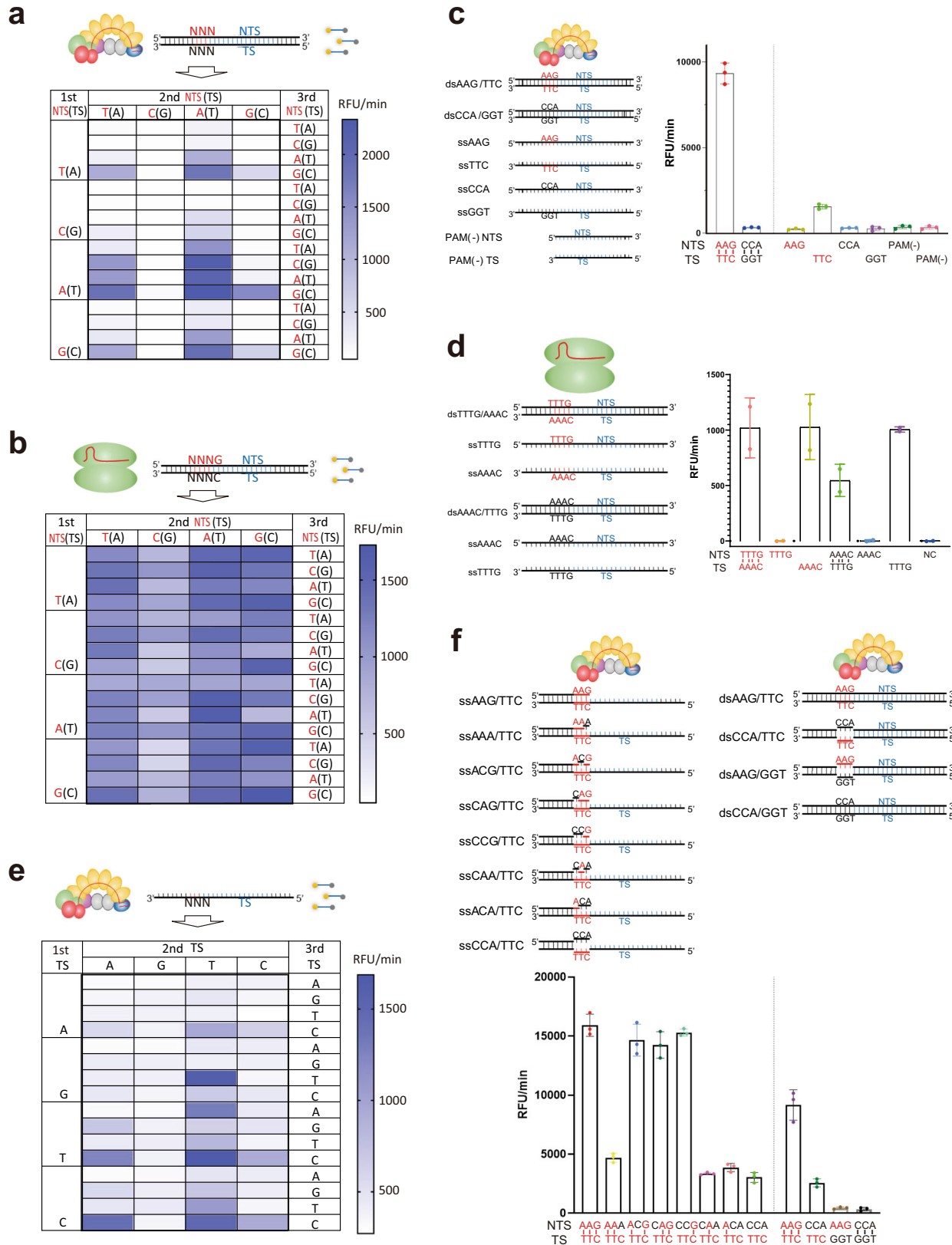

(3′-TTTG-5′) (Fig. 2d), as previously reported[34,35]. In contrast, EcoCas3 was partially activated by a TS with a PAM (3′-TTC-5′) but a TS with a nonPAM (3′-GGT-5′) prevented any activity (Fig. 2c). One potential explanation for why the PAM specificity was not observed in LbaCas12a is that the dsDNA targets are not fully double stranded, and some ssDNA contamination is sufficient to activate LbCas12a. We then tested

TS PAM specificity for all 64 possible target sites (Fig. 2e). The PAM specificities for ssDNA-activated collateral cleavage were similar to those of dsDNA-activated collateral cleavage (Fig. 2a), although the activity was mostly lower for ssDNA-activated cleavages, except for when the third nucleotide of the PAM was C, such as 3′-TAC-5′, 3′-AGC-5′, 3′-GTC-5′, 3′-GAC-5′, and 3′-GGC-5′, when the relative fluorescence

**Fig. 2 | Specificity of PAM recognition for collateral ssDNA cleavage by EcoCas3.** Screening of all 64 possible target sites containing each of the three-nucleotide PAM sequences for *trans* cleavage activity by EcoCas3 (**a**) and by LbaCas12a (**b**). The heat maps represent the RFU per min for collateral cleavage activity. **c** *Trans* ssDNA cleavage by a crRNA-complementary or non-complementary ssDNA (TS or NTS, respectively). EcoCas3/EcoCascade partially activated by TS ssDNA in a PAM-dependent manner (3′-TTC-5′ only). **d** LbaCas12a activated by TS ssDNA in a PAM-independent manner (both 3′-AAAC-5′ and 3′-TTTG-5′). **e** Screening of all 64 possible target sites containing each of the three-nucleotide PAM sequences for collateral cleavage activity by the TS ssDNA. **f** Collateral cleavage activated by crRNA-complementary ssDNA, where the PAM region was double stranded but the rest of the protospacer was single stranded. dsDNA-activated collateral cleavage is also shown to the right. Row data in the heatmaps of **a**, **b**, **e** is available in the Source data file. Data in **c**, **f** are presented for *n* = 3 independent measurements and mean value, error bars represent SD values. Data in d are presented for *n* = 2 independent experiments with central value.

was increased (Fig. 2e). This means the third nucleotide of the TS PAM is important for the collateral cleavage.

Base-pairing between the TS and NTS of the PAM leads to correct Cascade/crRNA binding of the NTS, accessibility of the EcoCas3 cleavage site, and degradation of the target DNA[24,49]. We observed that dsDNA containing an unpaired PAM between NTS-nonPAM (5′-CCA-3′) and TS-PAM (3′-TTC-5′) partially activated EcoCas3 for collateral ssDNA cleavage (Supplementary Fig. 10a). This is in contrast to previous reports of dsDNA with an unpaired PAM not showing any activity for target dsDNA degradation[24,49]. Screening of PAM base-pairing between each of the three nucleotides showed that base-pairing of the third nucleotide positively affected collateral cleavage activity, and that base-pairing of the first and second nucleotides additively increased the activity of the third nucleotide base-pairing (Supplementary Fig. 10b). Interestingly, the crRNA-complementary ssDNA, where the PAM region was double stranded but the rest of the protospacer was single stranded, displayed higher activity than either ssDNA- or dsDNA-activated collateral cleavage (Fig. 2f). This is probably because the protospacer ssDNA can easily bind to the complementary crRNA without unwinding the target dsDNA, which is followed by Cas3 recruitment and activation for ssDNA cleavage. Together, these results of PAM recognition specificity are mostly consistent with results from in vitro reconstitution[24,49] and of crystal structure analysis[18], except for the partial activity detected for collateral cleavage, in contrast to no activity for target DNA degradation by unpaired PAM recognition[24].

### EcoCas3 cleaves the NTS in *cis* followed by the TS in *trans* in a helicase-dependent manner

Complete R-loop formation by the Cascade/crRNA complex recruits the Cas3 helicase/nuclease, which repeatedly cleaves the NTS via the HD domain's single catalytic site[32,33]. It remains unknown how EcoCas3 cleaves the TS and progressively degrades the dsDNA substrate (Fig. 1a). Considering the collateral non-specific ssDNA cleavage in *trans*, we hypothesized that the TS can be cleaved in *trans*, following *cis* cleavage of the NTS after target dsDNA unwinding by the helicase properties of Cas3. To test this, we designed fluorescently labeled target dsDNA substrates, 5′-NTS-FAM, and 5′-TS-TAMRA, to visualize dsDNA cleavage by EcoCas3 (Supplementary Fig. 11a). In control experiments, SpCas9 cleaved both NTS and TS at 3–4 nucleotides upstream of the PAM site, as expected (Fig. 3a). In contrast, the highest peak of EcoCas3 cleavage was 10–11 nucleotides downstream of the PAM site on the NTS, while several peaks upstream of the PAM site demonstrated repetitive cleavage of the NTS. We also observed repetitive cleavage of dozens of nucleotides upstream of the TS PAM, which was likely reeled by EcoCas3 helicase activity and cleaved by its *trans* cleavage activity (Fig. 3a).

To confirm the NTS and TS cleavages mediated by nuclease/helicase activities of EcoCas3, we tested a dnCas3 HD domain mutant and a dhCas3 SF2 domain mutant in the dsDNA cleavage assay (Fig. 3b). The dnCas3 mutant cleaved neither NTS nor TS, indicating that the single catalytic domain of EcoCas3 plays a role in generating double-strand breaks (DSBs). Notably, the dhCas3 mutant cleaved the NTS, but not the TS, indicating that the dhCas3 mutant (S483A/T485A) works as an EcoCas3 Nickase (Fig. 3b). In ATP-free reaction buffer, the wild-type EcoCas3 and the dhCas3 mutant also cleaved the NTS, but

not the TS (Fig. 3c and Supplementary Fig. 11b). Given the assay's collateral cleavage results, where the dhCas3 mutant cleaved non-specific ssDNA (Fig. 1f), the helicase activity of EcoCas3 followed by reeling of TS is indispensable not only for repetitive *cis* cleavage of the NTS but also for *trans* cleavage of the reeled TS.

To further characterize *cis* and *trans* cleavage by EcoCas3, we compared 30 s (short) and 5 min (long) incubation times for the dsDNA cleavage assay. More prolonged incubation increased repetitive cleavage of the NTS in *cis* and the TS in *trans* (Supplementary Fig. 12a). We also observed that progressive *cis* and *trans* cleavages showed similar patterns in the repetitive experiments and the short and long incubation experiments, depending on the target DNA sequence, e.g., *hEMX1* and *mTyr* (Supplementary Fig. 12a). The sizes of many cleaved fragments were between 30 and 60 bps, which may be used for CRISPR adaptations as previously reported[12,50] (Supplementary Fig. 12b).

### Incomplete binding of EcoCascade to target DNA with tolerated mismatches elicits collateral ssDNA cleavage but not target dsDNA degradation

We previously reported that a single mismatch within the seed region markedly affected target DNA degradation in the EcoCascade/Cas3 system[11]. We, therefore, investigated the effect of mismatch for each nucleotide in the 32-nt spacer on collateral ssDNA cleavage activity. A single mismatch in the spacer region, even within the seed region (positions 1–8), resulted in little or no effect on collateral cleavage activity (Supplementary Fig. 13a, b). In the LbaCas12a system, 1–3 mismatches in the seed region also did not affect collateral cleavage activity (Supplementary Fig. 13c), consistent with previous reports[36,51]. Previous in vitro analysis revealed the effect of single mismatches in the target sequence, which slow the rate of R-loop formation and target-strand cleavage by Cas12a[52,53]. To investigate whether Cascade-binding and R-loop-formation are linked with collateral cleavage and target DNA degradation, we sought to characterize Cascade-target DNA binding kinetics using a Bio-layer interferometry (BLI) biosensor[54]. Corresponding to the collateral cleavage assay results (Fig. 2c), crRNA-complementary TS-ssDNA showed associations with EcoCascade but not with non-complementary NTS-ssDNA (Supplementary Fig. 14a and Supplementary Table 1). Notably, the crRNA-complementary TS-PAM (3′-TTC-5′) showed higher association than that of TS-nonPAM (3′-GGT-5′) or -PAMless (Supplementary Fig. 14a). Moreover, dsDNAs containing a paired PAM (5′-AAG-3′ − 3′-TTC-5′) showed the maximum EcoCascade-target DNA binding (Supplementary Fig. 14b and Supplementary Table 1), which corresponds to the results of the collateral cleavage assay (Fig. 2f). Unpaired PAM between TS-PAM (3′-TTC-5′) and NTS-nonPAM (5′-CCA-3′) indicated a lower association, and unpaired PAM between NTS-PAM (5′-AAG-3′) and TS-nonPAM (3′-GGT-5′) showed little association (Supplementary Fig. 14b). Taken together, BLI can provide solid information on the affinity and stability of interactions as previously reported[54].

To further investigate the relationship between the R-loop-formation and EcoCas3-mediated collateral ssDNA cleavage and dsDNA degradation, we assayed different length R-loop formations, from 0 to 32 nucleotides (n0, n6, n12, n18, n24, n30, and n32) (Fig. 3d). BLI revealed that crRNA-DNA hybridization with 0–12 base-pairs (n0, n6, and n12) including seed sequences did not show any association,

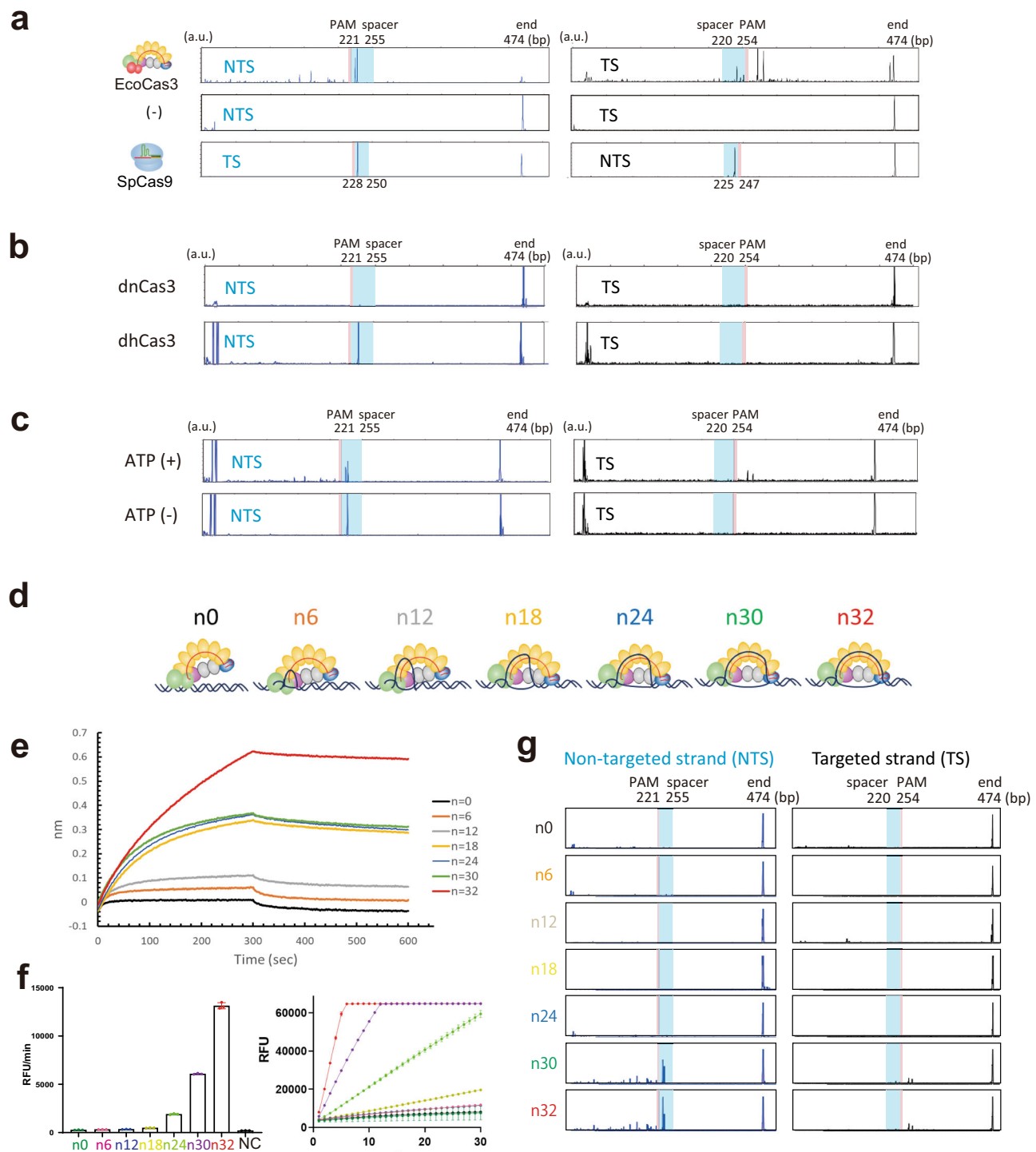

**Fig. 3 | Mechanistic insight into collateral ssDNA cleavage and target DNA degradation. a** Fluorescently-labeled target dsDNA substrates, 5′-NTS-FAM, and 5′-TS-TAMRA, to visualize dsDNA cleavage. EcoCas3 with EcoCascade RNPs cleaves at NTS nucleotides 10–11, downstream of the PAM site, with repetitive cleavage. The TS cleaved repetitively dozens of nucleotides upstream of the PAM. SpCas9 cleaves both NTS and TS at 3–4 nucleotides from the PAM. The *x*-axis represents the DNA fragment size in base pairs (bp), and the y-axis represents the arbitrary unit (a.u.). **b** The dnCas3 HD domain mutant and the dhCas3 SF2 domain mutant for the dsDNA cleavage assay. **c** The dsDNA cleavage assay by EcoCas3 in ATP (+) or ATP-free (−) reaction buffer. **d** Changing the size of the R-loop formation from 0 to 32

nucleotides by adding multiples of six nucleotides. **e** Measurement of EcoCascade-target DNA associations and dissociations in real-time using a bio-layer interferometry (BLI) biosensor (Octet RED 96 system). **f** In the collateral cleavage assay, n0–n12 base-pair hybridization did not show any cleavage activity, while n18–n32 R-loop formations increasingly promoted *trans* ssDNA cleavage activity. Data are presented for $n = 3$ independent measurements and mean value, error bars represent SD values. **g** In the dsDNA cleavage assay, the n0–n24 R-loop formation did not produce any cleavage of the NTS or TS. The 30–32 base-pair R-loop formations underwent repetitive cleavage on the NTS and the TS of the target dsDNA substrates. Source data are provided as a Source Data file.

while 18–30 base-pairs (n18, n24, and n30) produced a degree of association. Furthermore, a complete match for 32 base-pairs (n32) resulted in stable and emphatic Cascade binding, similar to locked R-loop formation reported previously[15,19,29] (Fig. 3e and Supplementary Table 1). In the collateral cleavage assay, n0, n6, and n12 did not show any cleavage activity, while n18–n32 R-loop formations increasingly promoted *trans* cleavage of ssDNA (Fig. 3f). In the dsDNA cleavage assay, n0–n24 did not show any cleavage of either NTS or TS DNA (Fig. 3g). This means that collateral cleavage does not need the nicking activity on the NTS (n18 and n24). Only the n30 and n32 sequences underwent repetitive cleavage on both the NTS and TS, and progressive cleavage of target dsDNA substrates (Fig. 3g). Taken together, these results show two Cascade binding modes. Intermediate R-loop formation by mismatches on the spacer sequences elicits collateral ssDNA cleavage. Complete R-loop formation with full crRNA-DNA hybridization leads to repetitive *cis* cleavage of the NTS with *trans* cleavage of the TS to degrade the target dsDNA substrate, as described in previous reports[15,19,29].

### Dynamic visualization of CRISPR interference: PAM search, nicking, and DSB

Cryo-EM and smFRET are not capable of visualizing how EcoCas3 degrades target dsDNA[12,31] (Fig. 1a). We therefore employed hs-AFM, which enables real-space and real-time observations of CRISPR-Cas3 reacting with target DNAs at the macromolecule level, as previously shown by visualizing CRISPR-Cas9 interference[55]. First, we visualized the binding of Cascade/crRNA to a target DNA, a 645 bp dsDNA containing a target spacer site flanked by a PAM (5′-AAG-3′) at 219 and 423 bp from the ends of the DNA fragment (Fig. 4a). We adsorbed the mixture of donor DNAs and EcoCascade RNPs onto a 3-aminopropyltrietoxysilane-mica surface (APTES-mica)[56]. As expected, hs-AFM demonstrated that EcoCascade RNP specifically bound to the expected target site in the dsDNA (Supplementary Fig. 15a–d). Notably, we frequently observed a typical DNA bend at the EcoCascade-RNP binding site (Fig. 4b). The DNA angle distribution around the bound EcoCascade RNP exhibited a large spread with a peak near 45°, while the DNA angle distribution around bare DNA showed an intrinsic gradual curve reflecting the persistence length of dsDNA, with or without Cas3[57,58] (Fig. 4c, d and Supplementary Fig. 15e, f). This DNA bending by a Cascade for stable R-loop formation was previously indicated by cryo-EM[15,18] and smFRET studies[16,59]. During the observation periods, the EcoCascade RNPs bound tightly to the target DNAs without dissociating, consistent with previous smFRET analyses[16,20,59].

We also observed target interrogation by the EcoCascade RNP on a mica-supported lipid bilayer (mica-SLB), which was indispensable to observe the dynamic events of EcoCascade RNPs, as previously used in the CRISPR-Cas9 system[55]. While the Cascades bound to the target site for more than 10 s in all cases, the duration of the non-target Cascade was mostly measured to be less than 1 s (Supplementary Fig. 16). We sometimes observed that the EcoCascade RNP ran from one end to the other through the target DNA (Fig. 4a, b and Videos 1 and 2), presumably searching for the right PAM site and spacer sequences, as shown by the smFRET studies[16,59].

Next, we injected EcoCas3 proteins after immobilizing the EcoCascade RNPs with the 645-bp target DNA in ATP-free reaction buffer on the mica-SLB to reproduce EcoCas3-mediated nicking at the target site. In this preparation, several single-strand break (SSB)-like DNAs at the Cascade binding site were observed (Fig. 4e and Video 3). We measured the DNA height because the preliminary experiments revealed a decrease in height at the nick site in artificially nicked DNA using Nb.BsrDI nicking endonucleases on APTES mica (Supplementary Fig. 17). Therefore, we observed pre-mixed EcoCas3-EcoCascade-dsDNA on APTES mica and measured the height of DNA for a more quantitative analysis. When removing the protein complex from

dsDNA by applying extensive force, we observed a nick-like shape, where the DNA chain remained connected, but the height of dsDNA appeared to be partially lower in ~60% of the complex (Supplementary Fig. 18a, b and Videos 3 and 4). In contrast, no nick structure was observed after mechanically removing EcoCascade from dsDNA without EcoCas3 protein (Supplementary Fig. 18c–e). These observations, therefore, demonstrate that EcoCas3 induces the nick as depicted in the in vitro experiments in Fig. 3.

In contrast, in ATP-containing reaction buffer, we detected many short DNA fragments after injection of EcoCas3 proteins (Supplementary Fig. 19). The DNA fragment lengths obtained varied widely but were commonly approximately 200 bps. Notably, we observed the EcoCas3-Cascade complex bound to the target site was repeatedly reeling the longer side of the DNA then releasing it, as previously reported[20] (Fig. 5a, b, Supplementary Fig. 20 and Videos 5–7). Finally, we captured the dynamic movements by which the EcoCas3-Cascade complex shortened the target DNA and cleaved it with a DSB after the reeling reaction, followed by release of the DNA from the EcoCas3-Cascade complex (Fig. 5b and Videos 6 and 7). These results clearly indicated that the DNA fragments can be cleaved upstream of the target site when they are shortened by reeling, rather than by Cas3 translocation.

## Discussion

Up until now, it has been unclear how a single HD nuclease domain in Cas3 can cause DSBs at target sites and long-range unidirectional deletions upstream of target sites[10,11,13,15,26,30,31]. We believe that this is the first report to use hs-AFM to capture the dynamic movements of CRISPR-Cas3 interference at the single molecule level. The hs-AFM results clarify that the EcoCascade/crRNA complex searches for and binds to target DNA, and recruited EcoCas3 bound to EcoCascade then reels and loops the target dsDNA, and subsequently cleaves it (Fig. 6). This is consistent with a reeling model in which Cas3 remains associated with Cascade to cleave ssDNA by a reeling mechanism[12]. However, it remained unknown how EcoCas3 cleaves the reeled TS and progressively degrades the dsDNA substrate (Fig. 1a). Our results from collateral ssDNA cleavage assays and dsDNA cleavage assays revealed that Cas3 repeatedly cleaves the NTS by helicase activity in *cis*. Simultaneously, the TS reeled by the helicase property of Cas3 can be cleaved by non-specific ssDNA cleavage activity in *trans*. The hs-AFM analysis also revealed that Cascade-bound Cas3 repeatedly reels and releases the target DNA upstream of the PAM site, followed by target degradation. (Supplementary Fig. 14). Although these results are inconsistent with a translocation model (Fig. 1a), Cas3 with Cas1 and Cas2 forming a primed acquisition complex may translocate in search of protospacers[20,60]. Despite hs-AFM being able to analyze regions of hundreds of bp, it is not feasible to visualize long-range dsDNA cleavages (in the order of kb) in vitro. Previous in vivo experiments with the CRISPR-Cas3 system[10,11] indicated unidirectional long-range deletions, where the spacer sequences and the PAM site remained in the absence of indel mutations and repetitive fragmented DNA deletions upstream of the PAM sites. The hs-AFM movement capture results support this CRISPR interference phenomenon, where the EcoCascade/Cas3 complex remains bound to the target site to repeat DNA degradation, which may expand the deletion size.

Previous studies revealed that type I CRISPR systems have two binding modes for target recognition called interference and priming[19,20,59,61] (Fig. 6). The low fidelity priming mode allows a whole range of mutated invaders to be detected and the priming process to be initiated[19,36,59,62,63]. Meanwhile, the high-fidelity interference mode ensures the destruction of perfectly matching targets to destroy foreign invaders without new spacer acquisition. Our results using the BLI biosensor show that partial R-loop formation (18–24 bp) enforces short-lived and unstable Cascade binding, whereas full R-loop formation (30 and 32 bp) provides locked and immobilized Cascade binding

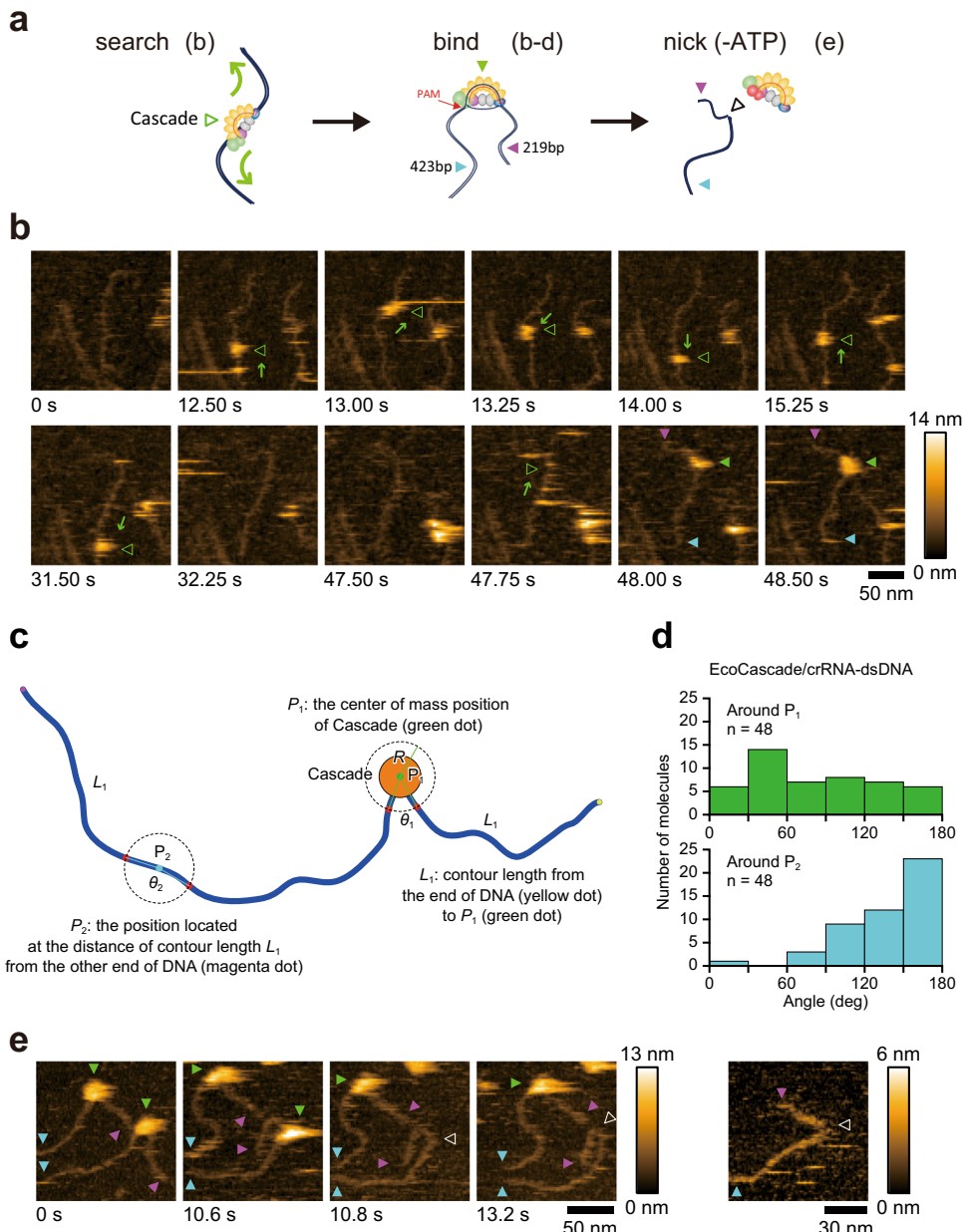

**Fig. 4 | Dynamic visualization of CRISPR-Cas3 binding and nicking by hs-AFM.**
**a** Schematic depictions of Cascade binding and Cas3-mediated nicking. **b** hs-AFM visualizes the EcoCascade RNP (green open triangle) searching for an appropriate PAM site from one end of the target DNA to the other (green arrows) (Video 1). EcoCascade binds to the target DNA (green closed triangle), a 674 bp dsDNA containing a target spacer site flanked by PAMs (5′-AAG-3′) at 219 bp and 423 bp from the ends of the DNA fragment (purple and blue triangle, respectively). Scan area: 200 × 200 nm² with 80 × 80 pixels; frame time: 0.25 s (line rate: 320 Hz). **c** Overview of strategies for measuring angles around Cascade. R:15 nm.
**d** Histograms of DNA angles around $P_1$ (top) and around $P_2$ (bottom) in EcoCascade-dsDNA. **e** Injection of EcoCas3 protein after immobilizing EcoCascade RNPs with the target DNA in ATP-free (−) reaction buffer to produce EcoCas3-mediated nicking (white triangle) at the target site (video 3). Scan area: 150 × 150 nm² (left 4 images) and 100 × 100 nm² (the most right image) with 80 × 80 pixels; frame time: 0.2 s (line rate: 400 Hz). The experiment of **b**, **e** was repeated twice independently with similar results.

to the target DNA (Fig. 3e). We also find that this partial Cascade binding can recruit Cas3 to mediate non-specific ssDNA cleavage in *trans*, but can also interrupt dsDNA cleavage in *cis* (Fig. 3f, g). Moreover, this collateral ssDNA cleavage tolerates mismatches within the spacer sequences (Supplementary Fig. 11a, b), in contrast to our previous findings with target dsDNA cleavage[11]. These findings suggest that the type I CRISPR system uses the collateral ssDNA cleavage for the priming process (Fig. 6). This is also supported by recently reported results showing that Cas12a has multiple nicking activities with tolerance of 4–8 mismatches within the PAM and spacer sequences in a natural role as an immune effector against rapidly evolving phages[36,51].

Our results also indicate that only stable Cascade binding can initiate nicking of the NTS and activate the ATP-dependent helicase property of Cas3, which reels and loops the TS for cleavage in the interference mode[15,19,29] (Fig. 6). It is still unclear what the critical step is that drives the Type I CRISPR interference mode. Previous cryo-EM studies reveal that full R-loop formation following conformational changes of Cascade triggers a flexible bulge in the NTS, enabling Cas3 nicking in this region[15,31]. In addition, the *trans* cleavage activities can be controlled by multiple steps including specific PAM recognition, R-loop formation-dependent conformational changes in Cascade, and recruitment of Cas3 and conformational changes in Cas3 itself, as previously suggested by several crystal structure or cryo-EM

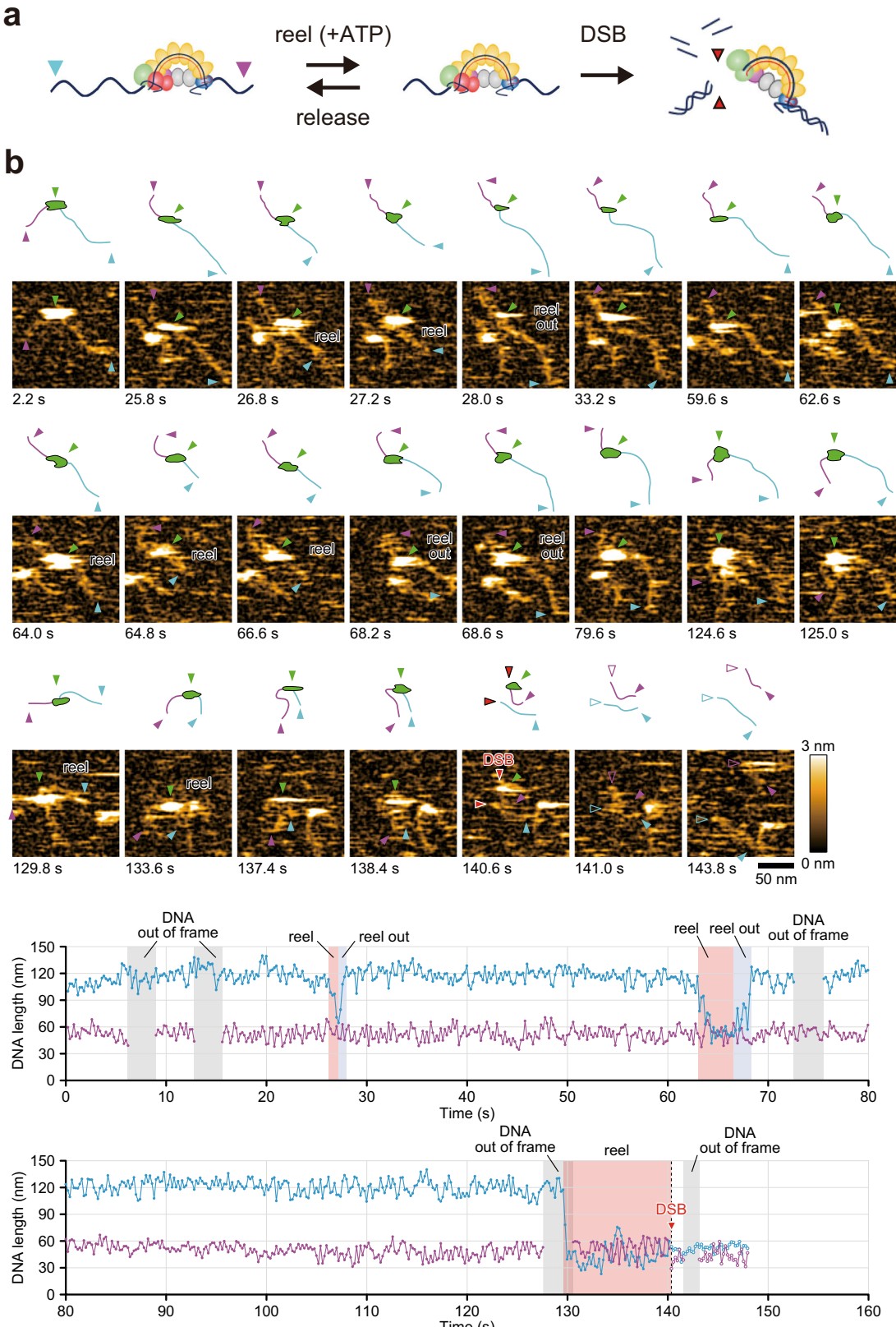

**Fig. 5 | Visualization of type I CRISPR interference. a** Schematic depictions of type I CRISPR interference. **b** In ATP (+) reaction buffer, the EcoCas3-Cascade complex repeatedly reels and releases the longer side of the DNA (blue arrows) and then cleaves it with a DSB (red arrows) (Video 5). Scan area: 200 × 200 nm² with 80 × 80 pixels; frame time: 0.2 s (line rate: 400 Hz). This experiment was repeated twice independently with similar results. Source data are provided as a Source Data file.

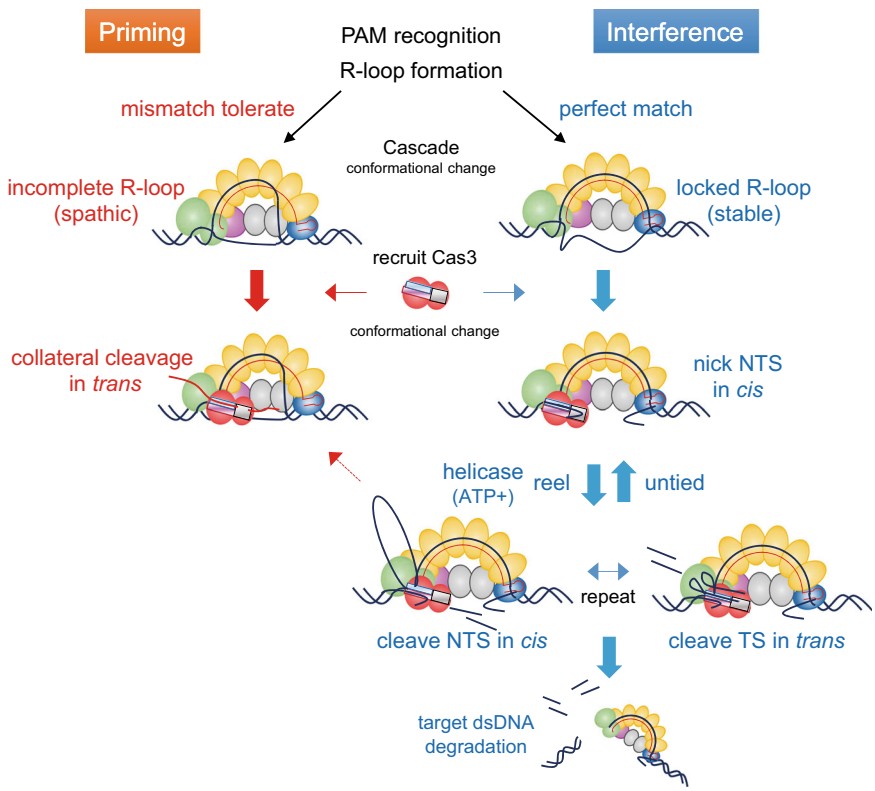

**Fig. 6 | Mechanism of type I CRISPR interference and priming.** Cascade binding to target DNA with tolerated mismatches elicits collateral ssDNA cleavage as a priming mode. Stable Cascade binding via a complete R-loop formation drives nicking of the non-target strand (NTS), followed by helicase-dependent unwinding of double-stranded DNA (dsDNA) upstream of the PAM site. Then, *trans* cleavage of the target strand (TS) combined with repetitive *cis* cleavage of the NTS degrades the target dsDNA substrate as an interference mode.

studies[15,17,18,21,22,27,64]. Importantly, Cas8 adopts a different conformation when bound to ssDNA for controlling Cas3 recruitment and activity, which correlates with the attenuation of interference and relative increase in priming activity upon target mutation[24,61]. Crystal structure[65] and cryo-EM[31] analysis also showed that Cas3 recruited to Cascade may be activated by guiding ssDNA to the HD domain by either of two routes, the bypass route or the helicase tunnel. Considering our findings and structural data[31,65], EcoCas3 may continuously cleave NTS via the helicase tunnel. Simultaneously, EcoCas3 cleaves the TS by collateral *trans* cleavage activity through the bypass route. By repeating *cis* NTS cleavage via the helicase tunnel route and *trans* TS cleavage via the bypass route, EcoCas3 may achieve phage plasmid degradation in *E. coli*[8] and large-scale genome editing in human cells[10,11] (Fig. 6).

The CRISPR-Cas3 system in the host bacteria potently degrades phage and viral DNA. If Cas3 is too powerful, it may have the potential for self-attack, from which the host must escape. EcoCas3 has a longer spacer sequence of 27 nucleotides compared with the 20 nucleotides of Cas9 or the 24 nucleotides of Cas12, which may increase the specificity for target recognition. EcoCas3 has maximal cleavage activity at 37 °C, although EcoCas3 protein is sensitive to temperature-dependent aggregation at 37 °C (Supplementary Fig. 3), which may also decrease self-attack. The specific PAM recognition by EcoCascade can also enable escape from self-attack (Fig. 2) because various CRISPR-Cas systems have a PAM system that distinguishes self from non-self[28]. Crystal structure analysis showed that Cas9 and Cas12 enable R-loop formation by first recognizing and unwinding the NTS-PAM[28]. Compared with Cas9, Cas12a loosely fits the PAM binding channel, allowing it to slightly open during suboptimal PAM binding[66]. The resulting loss of specific interactions between the PAM and the Cas12a channel can explain the observed higher *trans* cleavage

activities of Cas12a[66]. In the CRISPR-Cas3 system, EcoCas3 recruitment and binding to EcoCas8 depend on TS-PAM recognition. EcoCas8 binds to the third position of the TS-PAM and unwinds through recognition of the NTS-PAM[18], which may increase PAM specificity in collateral ssDNA cleavage (Fig. 2). Our experiments with the ssDNA where the PAM region was double stranded, but the rest of the protospacer was single stranded (Fig. 2f), suggested the proper Cas8 conformation with a double-stranded PAM allowing Cas3 recruitment, which may enable the partial ssDNA to fully activate *trans* cleavage. In the CRISPR-Cas3 system, the PAM plays an important role in self- and non-self-discrimination, and PAM recognition by Cas effectors is the initial step following the formation of an R-loop structure with the crRNA[18,28].

In conclusion, we found that the partial binding of EcoCascade to target DNA can elicit collateral non-specific ssDNA cleavage in the priming mode. With stable binding by a complete R-loop formation, the collateral ssDNA cleavage can be used for *trans* cleavage of the TS combined with repetitive *cis* cleavage of the NTS to degrade the target dsDNA substrate in the interference mode. These results provide a mechanistic insight into collateral ssDNA cleavage and target DNA cleavage by the CRISPR-Cas3 system, enabling further understanding of type I CRISPR priming and interference of foreign DNAs.

## Methods

### Expression and purification of EcoCas3 and EcoCascade/crRNA

We employed a method to express recombinant EcoCas3 at a low temperature using a baculovirus expression system. Briefly, we cloned an EcoCas3 cDNA with a octa-histidine tag and a six asparagine-histidine repeat tag into a pFastbac-1 plasmid (Thermo Fisher Scientific, Waltham, Massachusetts, USA) according to the manufacturer's instructions (Supplementary Fig. 2a). The TEV protease recognition

site was also inserted between the tags and EcoCas3 to enable tag removal. Self-ligation of the PCR product generated the mutant Cas3, such as H74A (dead nickase; dn) and S483A and T485A double mutant (dead helicase; dh) with PrimeSTAR MAX (Takara Bio, Kyoto, Japan). Coding sequences cloned in the plasmids are listed in Supplementary Data 1.

Expression of EcoCas3-tag fusion proteins in Sf9 cells. We infected Sf9 cells with baculovirus at a multiplicity of infection (MOI) of two at 28 °C for 24 h. Then, we changed the culture temperature to 20 °C four days after infection for protein expression. Sf9 cells were then collected and stored at −80 °C until use. The expressed EcoCas3-tag fusion proteins were purified using nickel affinity resin (Ni-NTA, Qiagen, Hilden, Düsseldorf, Germany). To remove tags, purified protein was digested with TEV protease and then further purified by size-exclusion chromatography using Superdex 200 Increase 10/300 GL (Thermo Fisher Scientific) in 0.2 M NaCl, 10% glycerol, 1 mM DTT, and 20 mM HEPES-Na (pH 7.0).

Cascade from *E. coli* and CRISPR RNA complex (EcoCascade/crRNA) was produced as described previously[24,41]. Briefly, we cloned EcoCas11 with a hexahistidine tag and HRV3C protease recognition site, EcoCascade operon, and pre-crRNA into pCDFDuet-1, pRSFDuet-1, and pACYCDuet-1 plasmids, respectively (Supplementary Fig. 2c). Sequences cloned in these plasmids are also listed in Supplementary Data 1. Then, we transformed JM109(DE3) with three plasmids to express EcoCascade/crRNA recombinant protein complex. Expressed recombinant EcoCascade-crRNA was purified using Ni-NTA resin. After removal of the hexahistidine tag by HRV3C protease, EcoCascade-crRNA was further purified by size-exclusion chromatography in 350 mM NaCl, 1 mM DTT, and 20 mM HEPES-Na (pH 7.0).

### Thermal stability assay of EcoCas3

Thermal stability was evaluated by nanoDSF using the Tycho NT.6 system (NanoTemper Technologies GmbH, München, Germany)[67]. Also, Thermal stability at a constant 37 °C was measured by a thermal shift assay using a Mx3000p real-time PCR instrument (Agilent technologies, Santa Clara, California, USA) and SYPRO orange (Thermo Fisher Scientific)[68].

### Single and double-stranded DNA preparation

To detect in vitro DNA cleavage activity of CRISPR-Cas3 proteins, targeted sequences of *EMX1* with PAM variants (5′-AAG-3′ or 5′-CCA-3′) were cloned into a pCR4Blunt-TOPO plasmid vector (Thermo Fisher Scientific) according to the manufacturer's protocol. For collateral DNA cleavage assays, 60 bp activator fragments of *hEMX1* and *mTyr* (which included a target site) were designed and purchased. Targeted sequences for CRISPR-Cas3, CRISPR-Cas12a, and CRISPR-Cas9 are listed in Supplementary Data 2. PAM sequence variants and targeted sequence variants were also designed to examine collateral ssDNA cleavage activity. Biotin-labeled fragments were also purchased for protein-DNA interaction analysis. For fragment analysis, fluorescence-labeled primers were designed and the DNA fragment amplified from genomic DNA of HEK293T cells using Gflex DNA polymerase (Takarabio). Amplicons were purified using NucleoSpin Gel and a PCR Clean-up kit (Takara-bio) according to the manufacturer's protocols. A DNA fragment for hs-AFM was also amplified with non-labeled primers. All sequences of primers and donor DNAs are listed in Supplementary Table 3 and Supplementary Data 2, respectively.

### In vitro DNA cleavage activity

To analyze DNA cleavage activity, 1.6 nM of plasmid with or without targeted sequences were added to 115 nM EcoCascade-crRNA complex, 250 nM EcoCas3, and 2.5 mM ATP in CRISPR-Cas3 working buffer (60 mM KCl, 10 mM MgCl₂, 10 µM CoCl₂, 5 mM HEPES-KOH, pH 7.5), as previously described[34,43,47]. After incubation at 37 °C, samples were detected by either electrophoresis or with the MultiNa microchip electrophoresis system and the DNA-12,000 kit (Shimadzu, Kyoto, Japan).

### Reporter assay for DNA and RNA cleavage

To characterize Cas3 collateral nucleic acid cleavage activities, 50 nM DNA activator templates were added to 100 nM EcoCascade-crRNA complex, 250 nM EcoCas3 and 2.5 mM ATP in CRISPR-Cas3 working buffer (60 mM KCl, 10 mM MgCl₂, 10 µM CoCl₂, 5 mM HEPES-KOH, pH 7.5). We used the DNase Alert kit (Integrated DNA Technologies, Coralville, IA USA) and the RNase Alert kit (Integrated DNA Technologies) for detecting DNase and RNase activity, respectively. To measure the ssDNA cleavage activity, we used the qPCR reporter probe for GAPDH (the sequence is listed in Supplementary Table 3) at 125 nM. Cleavage-related change in fluorescence signal of the probe was measured every 30 s for 60 min under incubation at 37 °C using a Real-time PCR system (Bio-Rad Laboratories, Hercules, California, USA). Alternatively, M13mp18 single-stranded DNA (New England Biolabs, Ipswich, Massachusetts, USA) or pBluescript plasmid were added and incubated at 37 °C. Samples were then electrophoresed on an agarose gel.

### DNA fragment analysis

To analyze CRISPR DNA cleavage patterns in vitro, 16 nM DNA fragments amplified from HEK293 genomic DNA were added to 160 nM EcoCascade-crRNA complex, 400 nM EcoCas3 and 2 mM ATP in CRISPR-Cas3 working buffer (60 mM KCl, 10 mM MgCl₂, 10 µM CoCl₂, 5 mM HEPES-KOH pH 7.5). After incubation at 37 °C, DNA samples were purified by ethanol precipitation. The length of DNA in samples was measured using GeneScan 600 LIZ dye Size Standard (Thermo Fisher Scientific) via a G5 dye set filter. All data were analyzed using PeakScanner software (Thermo Fisher Scientific).

### Protein-DNA interaction assay

The evaluation of binding properties between EMX-EcoCascade (analyte) and target DNAs (ligands) was performed by BLI using the Octet RED 96 system (ForteBio, Sartorius BioAnalytical Instruments, Fremont, California, USA). All ligands were biotinylated (20 µM) and immobilized on streptavidin biosensors. Kinetic titration series were performed in interaction buffer (PBS with 0.01% Tween 20, 0.02% BSA). Analyte concentration was 20 µM in the interaction buffer. The association and dissociation times were both 300 s to measure the interaction between ligands and analyte. These raw data were analyzed using ForteBio analysis software. The binding sensorgram was locally fitted to a 1:1 Langmuir binding model with mass transport limitation. Sequences for the donor DNA fragments were listed in Supplementary Data 2.

### High-speed atomic force microscopy (hs-AFM) setup

hs-AFM imaging was performed in solution using a laboratory-built hs-AFM setup as described previously[55]. We used small cantilevers (BLAC10DS-A2, Olympus, Tokyo, Japan) with a nominal spring constant of 0.1 N/m, resonance frequency of ~0.5 MHz, and a quality factor of ~1.5 in the buffer. The probe tip was grown on the original tip of a cantilever through electron beam deposition using ferrocene and was further sharpened using a radio-frequency plasma etcher (Tergeo, PIE Scientific LLC, USA) under an argon gas atmosphere (Direct mode, 10 sccm, and 20 W for 1.5 min). A glass sample stage (diameter, 2 mm; height, 2 mm, Japan Cell, Tokyo, Japan) with a thin mica disc (1.5 mm in diameter and ~0.05 mm thick, Furuuchi Chemical Corporation, Tokyo, Japan) glued to the top by epoxy was attached the top of a Z-scanner with a drop of nail polish.

Engagement of the probe with the surface before scanning was performed using a stepping motor[55]. The speed of engagement was ~0.7 µm/s when the distance between the probe and surface was more than 10 µm. After that, the speed was slowed to ~0.3 µm/s. During engagement, the feedback control for the cantilever's oscillation

amplitude was applied with a moderate feedback gain, in which the cantilever's free oscillation amplitude, $A_0$, and setpoint amplitude, $A_{sp}$, were set at ∼2 nm and ∼0.95 × $A_0$, respectively. This setting was important to avoid the probe from making a strong contact with the surface. After engagement, $A_0$ and $A_{sp}$ were set at 1−2 nm and ∼0.9 × $A_0$, respectively. While taking AFM images, the feedback gain, mainly the integral gain and supportively the proportional gain, was adjusted until clear images appeared. When a strong tapping force was required, $A_{sp}$ was lowered to ∼0.3 × $A_0$.

## Observation by hs-AFM

To observe the reaction intermediates of EcoCascade/crRNA and dsDNA before and after addition of EcoCas3 and/or ATP at high spatial resolution, we used 3-aminopropyltriethoxysilane treated mica (APTES-mica) as described previously[55]. The premix complex of Eco-Cascade/crRNA and dsDNA was prepared in observation buffer (5 mM HEPES-KOH, pH 7.5, 30 mM KCl, 1 mM $MgCl_2$, 2 μM $CoCl_2$, 10% glycerol) at room temperature (∼25 °C) using 1.5 nM EcoCascase/crRNA and 0.5 nM dsDNA. The premix complex of EcoCascade/crRNA, dsDNA and EcoCas3 was prepared in observation buffer using 75 nM Eco-Cascade/crRNA, 25 nM dsDNA, and 150 nM EcoCas3, and incubated at 37 °C for 30 min, after which the solution was diluted 50 times with observation buffer. The reaction intermediates of EcoCascade/crRNA and dsDNA after the addition of EcoCas3 and ATP were prepared in observation buffer using 75 nM EcoCascade/crRNA, 25 nM dsDNA, 150 nM EcoCas3 and 1 mM ATP, and incubated at 37 °C for 5 min, after which the solution was immediately placed on ice to stop the reaction. Just before use, the solution of the reaction intermediates was diluted 50 times with observation buffer. The artificially nicked dsDNA was diluted to 1 nM with Milli-Q water. A drop (2 μl) of either sample solution was deposited on the APTES-mica. Three minutes later the sample surface was rinsed with 20 μl observation buffer. We then immersed the sample stage in a liquid cell containing about 60 μl observation buffer[69,70], and hs-AFM imaging was performed at room temperature (∼25 °C). The addition of 10% glycerol in the buffers helped to prevent deterioration of the probe tip, but not completely.

To observe dynamic behaviors of EcoCascade and EcoCas3 on dsDNA, we used a mica-supported lipid bilayer (mica-SLB) as a sample substrate[55,69,71]. To observe EcoCascade binding to a target site, a lipid composition of 90:5:5 (w/w) DPPC:DPTAP:biotin-cap-DPPE was used. The liposome solution dissolved in Milli-Q water at 2 mg/ml was prepared as described previously[55] and stored as 50 μl aliquots at −20 °C before use. After thawing an aliquot, the liposome solution was diluted with the equivalent volume of 10 mM $MgCl_2$ and sonicated with a bath sonicator (AUC-06L, AS-One, Osaka, Japan) to obtain small unilamellar vesicles. A drop (2 μl) of the sonicated liposome solution was deposited on the surface of freshly cleaved mica for more than 5 min at room temperature (24−26 °C), during which the stage was covered with a humid cap to avoid surface drying. The sonicated liposome solution could be stored at −20 °C and used repeatedly by applying the sonication. Before deposition of the samples, the surface of the sample stage was rinsed with a large amount of Milli-Q water (∼20 μl × five times) to remove excess liposomes and lipid bilayers. We deposited 2 μl dsDNA amplicon diluted to 5 nM with Milli-Q water on the mica-SLB. Three minutes later the sample surface was rinsed with 20 μl observation buffer to remove the unattached dsDNA. We then immersed the sample stage in a liquid cell containing about 60 μl observation buffer, and hs-AFM imaging was performed in a room heated to 30 °C. We added a drop (∼6 μl) of EcoCascade diluted with observation buffer, which was equilibrated to ∼30 °C, to the liquid cell during the hs-AFM observations, resulting in a final concentration of ∼20 nM.

To observe DNA reeling and DSB generation by EcoCas3, the lipid composition used was 80:10:10 (w/w) DPPC:DPTAP:biotin-cap-DPPE. Increasing the ratio of biotin-cap-DPPE improved the efficiency of the reeling and the DSB reaction induced by EcoCas3. A drop (2 μl) of EcoCascade-DNA pre-assembled with 20 nM EcoCascade in observation buffer was placed on the mica-SLB together with 2 nM DNA at 37 °C for 5 min. The sample surface was rinsed with 20 μl observation buffer and then imaged with hs-AFM, with a head temperature controlled at ∼37 °C, which is a suitable temperature for DSBs, using a thermostatic cover. During the hs-AFM observations, a drop (∼6 μl) of EcoCas3 and ATP mixture diluted with observation buffer to a final concentration of ∼100 nM and ∼2 mM, respectively, was equilibrated to ∼37 °C and added to the liquid cell.

Under the above observation conditions, at least one target molecule (i.e., the complex of EcoCascade/crRNA and dsDNA with or without EcoCas3, with dsDNA generated by the DSBs reaction or artificially nicked dsDNA) was observed in an observation area of 500 × 500 $nm^2$, such that the target molecule can be easily encountered. The reproducibility of the AFM observations was confirmed by performing at least three independent experiments. As DPTAP has a positively charged head group, addition of DPTAP facilitated immobilization of negatively charged DNA on mica-SLB. As biotin-cap-DPPE has a bulky head group, addition of biotin-cap-DPPE induced sub-nanometer roughness on mica-SLB. This increased the mobility of dsDNA on the surface and sometimes allowed the observation of binding events of proteins that surround dsDNA as previously seen for CRISPR-Cas9[55]. Primers for the donor DNA amplicons are listed in Supplementary Table 3.

## Reporting summary

Further information on research design is available in the Nature Research Reporting Summary linked to this article.

## Data availability

The data that support this study are available from the corresponding author upon reasonable request. Source data are provided with this paper.

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

## Acknowledgements

We thank T. Omoto, and S. Kobori at Osaka University, M. Hoshi, K. Mochizuki, and A. Fukui at Tokyo University, and S. Saji, S. Yamamoto, and N. Godai at the RIKEN SPring-8 Center for their technical assistance and Drs. T. Ando and T. Watanabe-Nakayama for technical support with hs-AFM. We also thank Professor Jacob Corn at ETH Zurich for scientific advice and Jeremy Allen, PhD from Edanz Group (https://en-author-services.edanz.com/ac) for editing a draft of this manuscript. This project was supported by JSPS KAKENHI Grant Number 18H03974 (T.M.), 19K16025 (K.Y.), 19KK0401 (K.Y.) and 20H00327 (N.K.), and JST-CREST (JPMJCR1762 to N.K.). Support also came from the Platform Project for Supporting Drug Discovery and Life Science Research [Basis for Supporting Innovative Drug Discovery and Life Science Research (BINDS)] from AMED under Grant Number JP20am0101070 (support numbers 1251 and 2463).

## Author contributions

K.Y. and S.S. designed and performed most of the experiments, analyzed the data with assistance from Y.K. and Y.Y. K.T., M.O., and M.Y. prepared and characterized the CRISPR-Cas proteins. N.K., K.U., K.Y., and K.T. performed the hs-AFM experiments. T.M. conceived and supervised the study, and wrote the manuscript with editorial contributions from all authors.

## Competing interests

S.S. and Y.K. are employees of C4U Corporation. K.Y., K.T., and T.M. are scientific advisors to C4U Corporation, which does not affect any result and conclusion reported in the paper. The other authors declare no competing interests.
