## [Peer Review File · Nature Communications]

REVIEWER COMMENTS

Reviewer #1 (Remarks to the Author):

In this manuscript, Yoshimi et al. describe a new activity of Cas3, the endonuclease effector of the type I-E CRISPR-Cas system. Type I-E is among the best studied CRISPR-Cas system due to its presence in *E. coli*. Although extensive mechanistic and structural studies have been performed for this system, several questions remain. In particular, this study addresses questions related to how Cas3 is able to degrade both strands of the DNA target, when only one strand (the non-target strand) is known to be threaded in the nuclease active site. The major new finding of this paper is that Cas3 has trans collateral cleavage activity when activated by Cascade, similar to Cas12a collateral cleavage upon target activation. The authors demonstrate this activity very convincingly, and also perform interesting dsDNA cleavage assays that show how the target strand gets cleaved by Cas3 in trans. The authors also perform high-speed AFM experiments that allow visualization of the interference process. Although the videos of the AFM experiments are difficult to follow for the untrained eye, the authors provide snapshots in the static figures that help the reader follow the process.

The AFM experiments are an interesting inclusion, but the most important finding is the trans cleavage by Cas3. This cleavage activity adds to a growing list of activated nickases that have been reported for Cas proteins. Given the recent development of CRISPR-Cas3 genome editing as well as Cas12a, Cas13 and Csm6-based diagnostics, it is likely that this finding will generate interest not only among the CRISPR community, but also in the applications fields. I have a few concerns about the way data is presented (listed below), but otherwise I am supportive of publication.

1. Many of the results are recapitulations of well-known Cascade-Cas3 activities. The authors could consider shortening the Results section significantly by moving description of Cas3 purification to the Methods and reducing discussion of the effects of mismatches on target binding. The data shown in Figs. 1b-c are controls showing that the system the authors have reconstituted works similar to prior studies, and could be described more briefly.

2. The collateral cleavage with ssDNA targets show interesting differences between Cas3 and Cas12a. A likely explanation for this difference is due to inefficient recruitment of Cas3 due to Cas8 conformation. It is known that Cas8 adopts a different conformation when bound to ssDNA (see refs 24 and 59), which is thought to reduce Cas3 recruitment. This could be added to the Results or Discussion section. On the other hand, it is unclear why the PAM sequence makes a difference in this case. It has previously been thought that PAM must be double-stranded for its recognition, consistent with its recognition in crystal and cryo-EM structures. It might be interesting to do an experiment with the same ssDNA where the PAM region is double stranded but the rest of the protospacer is single stranded (i.e. using a partially double-stranded DNA in which only the PAM and upstream region are double-stranded). With a double-

stranded PAM allowing proper Cas8 conformation, Cas3 recruitment may be enabled and the partial ssDNA may fully activated trans cleavage.

3. It is surprising that LbaCas12a is so permissive of PAM sequences for collateral cleavage. This is, to my knowledge, inconsistent with prior publications (e.g. Chen et al. 2018), which show the expected PAM-dependence for Cas12a dsDNA targeting. Do the authors have an explanation for this?

4. In the cleavage experiments with the NTS and TS labeled, it would be helpful to not only label the location of the PAM, but also the location of the full length DNA and the location of the protospacer in each chromatogram. The protospacer could be indicated by shading or a box. It is very difficult to read the numbers on the x-axis of these chromatograms, so a sense of scale is missing from the figure.

5. Many figures have very small font sizes, making it hard to read. Please adjust all font sizes to at least 6 point font.

6. On 389-390 what is meant by “which carry small mutations”? Do you mean that these create DSBs rather than long range cleavage events? Please clarify, or potentially remove this sentence, which seems speculative.

7. There are several figures where Cascade is spelled Casscade (Fig 1b, 1c, Extended data figure 2c)

8. In general, the references could be updated a bit to better reflect the literature. I urge the authors to go through the introduction and discussion and make sure the correct and comprehensive references are cited to support individual statements. A few specific examples are listed below.

a. On line 59-60, the authors cite a number of references, including 2 papers reporting the first crystal structures of Cascade. Another paper reporting the same structure (PMID 25118175) was published at the same time but was omitted from these references. Please add this as well.

b. On line 337, these two papers used single molecule fluorescence assays (DNA curtains assays) but not smFRET. Similarly, on line 346, only one of these papers (57) was an smFRET study. Please update these sentences.

c. The sentence at lines 357-359 seems redundant with other parts of the paragraph. This could be mentioned when priming is first mentioned at the beginning of the paragraph. It would also be appropriate to cite the first papers to demonstrate priming in vivo (Datsenko et al, 2012 and Swarts et al, 2012)

d. Line 372: not all of these studies are cryo-EM studies, some of them report crystal structures.

e. Line 381: Only ref 66 reported a crystal structure of Cas3, ref 30 reported a cryo-EM structure of Cascade-Cas3 complex.

Reviewer #2 (Remarks to the Author):

1. Conflicts of interest: Several of the authors are from a company. The interest of the company in this work should be explained so that the readers can make their own judgements about potential biases.

2. General HSAFM methods description: The authors cite reference 55 for some of the key methodological details and it would be much more helpful for them to include the relevant information in the manuscript or supplementary info, versus expecting the reviewer to search through older references.

Details of the scanning parameters used in this study need to be added - both for the APTES surface experiments and the two different lipid bilayer experiments. Specifically, what was the probe linear velocity across the surface? What was the scan dimension and frequency in the fast scan and slow scan axes? Engagement of the probe with the surface before scanning is a very important part of the procedure. This needs to be described. What was the velocity profile as the probe was moved vertically to engage the surface? What force conditions or other parameters were used to indicate a successful engage condition? This is important to know, because in some cases, these parameters can be different from the steady-state feedback set point conditions used during imaging.

The authors need to include some comments about the difficulty and reproducibility of the imaging. How many experiments are required to get a successful result? How many different regions are interrogated on the surface during a single experiment? How many repeats were conducted? Was the DNA evenly distributed on the surface or did the operator need to hunt for a suitable region to image before proceeding? Is there a time after adding the protein complexes that the system is no longer viable or good data can no longer be recorded? i.e. how fast does the experiment need to be conducted?

Many of the sample prep details that would be required for other scientists to reproduce these experiments are missing. This is an important deficiency because AFM experiments often hinge on successful sample prep as much as, or more than, on instrument operation. For example, a lipid supported bilayer was used as a substrate for some of the observations. The procedures for preparation of the bilayer and application of the bilayer to the mica surface need to be described in sufficient detail

such that they could be reproduced by others. The solution conditions for the DNA amplicons applied to the bilayer also needs to be described, as does the volume of liquid applied and the surface area of the substrate that is covered. Is the DNA solution removed or otherwise discarded before the wash step? What is the estimated final surface coverage of the DNA? i.e. how many molecules are retained on the surface per square micron. This is relevant because the molar ratio of protein complex to DNA strand needs to be estimated so that kinetics can be inferred, as well as for practical purposes of reproducing the experiment.

For the lipid bilayer experiments, is the observation buffer the same as described in the preceding paragraph, for the APTES surfaces? Is the mica substrate fixed to the scan stage during this procedure, or is it fixed after DNA application and washing? Details of the liquid cell design and geometry should be included, as well as a schematic of how the cell and the HSAFM scan stage are assembled together. If already described in a previous publication, a cursory description and reference should be included in this manuscript. The EcoCascade was added at some point after the sample was in the instrument. What were the solution conditions for the EcoCascade sample added? The room was heated to 30C -- was the fluid cell and the sample liquid equilibrated at the same temperature? If so, how long was the EcoCascade sample maintained at 30C after it was taken from cold storage and introduced to the cell?

Why was a different bilayer preparation used for observation of EcoCascade binding versus Cas3 reeling and double-strand break behavior? In the reeling and ds break experiments, the sample was maintained at a temperature of 37C -- why was the temperature different from the earlier experiments (30C). Is this change inconsequential, or relevant to the procedure?

3. HSAFM data. In general, the quality of the HSAFM images is poor, to the point that many of the claims I cannot evaluate. Furthermore, in these kinds of observational experiments, many events and the associated statistics need to be presented before drawing conclusions. Drawing conclusions from observation of one or two molecules when there are likely millions on the surface in a single experiment and multiple repeats could be conducted, is not methodologically sound, in my opinion. For these reasons, I find that this group of experiments detracts from, rather than adds to, the quality of the manuscript.

Figure 4a. This is a cartoon, not an observation. The manuscript (line 291..) describes it as an observation, which is confusing.

Figure 4b and video 1 and video 2. Video 1 and the frames abstracted from it are fairly clear. Video 2 claims to show a bound complex and R-loop architecture. However, video 2 is too blurry for this reviewer to interpret. It appears that the protein complex binds to a strand but does not move.

Importantly, the manuscript indicates (line 297..) that many EcoCascade RNPs formed stable complexes and bound to the target site. However, it is not clear what data (which videos and images) are used to make this conclusion. Also claimed is that the complexes bound tightly to the target DNAs without dissociation. In video 1, there appears to be a complex of proteins moving along the DNA, while also occasionally desorbing and re-adsorbing. It is not clear if this is a complex or a partial complex.

Figure 4c and video 4. This video and associated frames appear to show two protein complexes bound to two different DNA strands at roughly the same relative position from the end from each strand. The relative position corresponds to where the expected target site would be, though there is nothing to ensure that this is the case for these molecules because the two ends are not individually distinguishable. How many different times was this behavior observed? Can we be sure that this is not a random observation? The video/figure also claims to show the formation of a nick site, but this entirely based on a kink structure in the DNA backbone, not actual observation of the nick itself. Since there are many possible causes of a kink, including a nick, this observation is not particularly compelling without some other sort of identification, such as enzymatic labeling of the actual nick.

Figure 4d and video 6. The image contrast and noise character of these images are too poor for this reviewer to interpret them. Since discussion of 'reeling' observation is based on this data, I cannot evaluate the claims in the manuscript.

Reviewer #3 (Remarks to the Author):

Yoshimi et al. have elucidated the molecular mechanism of CRISPR interference by the CRISPR-Cas3 system by employing a variety of biochemical and biophysical techniques. The most important aspect of this work is the finding that the binding of type I CRISPR Cas effectors to target dsDNA has two modes depending on the binding strength, switching between cleavage of the collateral ssDNA and the target dsDNA and proposing a mechanistic model for type I CRISPR priming and interference against a foreign DNA. Direct observations of dynamics by hs-AFM are really interesting because of the first direct observation of the sequence of events from DNA binding to dsDNA cleavage in the Cas3 system at the single molecule level, but it does not seem to be directly related to the main claims of this manuscript. In fact, it is a bit strange to discuss hs-AFM data at the beginning of discussion when it is not even mentioned in the conclusion. Rather, it would be more effective to demonstrate the use of hs-AFM by imaging the strength of the binding between EcoCascade and the target DNA and the fluctuation of the binding, as in the experiment shown in Fig.3d. Since the conclusions of the manuscript itself are intriguing and provide mechanistic insights of CRISPR interference, it is worthy of eventual publication in

Nature Communications. However, the technical terms that are key to understanding the overall argument are not well defined and the explanations are very difficult to understand. In particular, the definition of trans- and cis-cleavage was not clear to the reviewer who is not an expert of CRISPR systems, even though the important point of this paper cannot be understood without knowing it. It should be explained first so that non-specialist readers can understand it even if it is obvious to experts. Also, data is not properly selected, making the manuscript redundant and difficult to read. There is a lack of quantitative analysis regarding HS-AFM data, which is too intuitive. That's important, but the discussion should be based on quantitative analysis whenever possible.

Below are some questions and comments that should be considered for revising the manuscript.

1. First of all, please explain the meaning of trans- and cis-cleavage with an illustration in the introduction. Though it may be obvious to experts, it is quite hard to read the manuscript without understanding the meaning of terms.

2. In Fig. 1d, the authors investigated the effect of divalent ions on non-specific cleavage of ssDNA, but I don't understand what the authors are trying to claim with this result. Is there any effect of divalent ions on ssDNA cleavage in the presence of the target dsDNA?

3. page 8, lines 161-163, "In ATP-free reaction buffer (-), the collateral activity of the EcoCas3 protein was at the same level as that of wild-type EcoCas3 and the dhCas3 mutant in ATP (+) buffer (Fig. 1f)."

The authors concluded that the collateral ssDNA cleavage activity of EcoCas3 in ATP-free buffer is the same level as that in the presence of ATP. However, it is difficult to conclude that to me that they are at the same level, simply looking at this graph. The cleavage activity appears to be higher in the ATP-free condition. The authors should clearly explain the basis for concluding that the activity is at the same level. If the activity is increased in ATP-free, the possible cause should be mentioned.

4. page 8, line 163-165, "Together, these results indicate that the nuclease and helicase activities of EcoCas3 are required for target DNA degradation, but only the nuclease activity is required for collateral cleavage."

For dnCas3, does it have the helicase activity of dsDNA without the cleavage activity? I mean, does the unwinding of dsDNA occur by dnCas3?

5. page 8, lines 177-185, "In contrast, LBaCas12a..."

I don't understand why the authors need to show the Cas12a results in this manuscript even though the results are the same as previous studies. Shouldn't they just focus on the Cas3 results?

6. pages 8-9, "PAM recognition is a prerequisite for collateral ssDNA cleavage by Cas3 but not Cas12a."

The experimental results shown in Figs. 2a-e are based on the NTS PAM sequence, but Figs. 2c and d are explained by the TS sequence, which is very complicated. Fig. 2a-e should also be shown with the PAM sequence of the TS.

7. page 9, lines 190-192, "..., except for when the third nucleotide of the PAM was C, such as TAC, AGC, GTC, GAC, and GGC,..."

What are the implications of these results? Are there any possible explanations for the results in the model description?

8. page 9, lines 196-198, "This is not consistent with a previous report that showed dsDNA with an unpaired PAM did not activate EcoCas3 to degrade target DNA substrates."

Fig. 2f shows the result of collateral ssDNA cleavage activity, which is not different from the degradation of the target DNA. I don't understand what the authors say "not consistent".

9. I have no idea how to look at the data shown in Fig. 3a-c, and there is no explanation even in the method section. There is no description of the meaning of the horizontal and vertical axes of the graph, and even the meaning of the colored dashed lines in the figures. I understand what the authors mention from the context, but it should be explained properly so that everyone can understand.

10. page 10, lines 227-229, "Notably, the dhCas3 mutant cleaved the NTS in cis, but not the TS in trans (Fig. 3b), which was not consistent with the assay's collateral cleavage results where the dhCas3 mutant cleaved non-specific ssDNA in trans (Fig. 1f)."

I don't understand the meaning of "the NTS in cis" and "the TS in trans" in this sentence. Why not simply the NTS and the TS? Same as the very first question, it should be explained clearly since it seems the key point of this manuscript.

11. page 11, lines 238-242, "We also observed that progressive cis and trans cleavages showed similar patterns in the repetitive experiments and the short and long incubation experiments, depending on the target DNA sequence (Fig. 3 and Extended data Fig. 10a). The sizes of many cleaved fragments were between 30–60 bps, which may be used for CRISPR adaptations as previously reported (Fig. 3 and Extended data Fig. 10b)."

I have no idea what Fig.3 in these sentences refers to or what it is talking about. Is the figure number correct?

12. page 11, lines 251-252, " In the LbaCas12a system, 1–3 mismatches in the seed region also did not affect collateral cleavage activity (Extended data Fig. 11c), consistent with previous reports"

Is it really necessary to show the data even though it is only to confirm the previous results?

13. page. 13, lines 291-294, "First, we visualized the binding of Cascade/crRNA to a target DNA, a 645-bp dsDNA containing a target spacer site flanked by a PAM (AAG) at 219-bp and 423-bp from the ends of the DNA fragment (Fig. 4a). We then adsorbed the mixture of donor DNAs and EcoCascade RNPs onto a 3-aminopropyl-triethoxy silane-mica surface (APTES-mica)."

The two sentences should explain the conditions of different experiments. This way of writing misleads readers into thinking that it is a continuous experiment. The second condition should be mentioned when showing the corresponding data.

14. page 13, lines 297-298, "We also found that many EcoCascade RNPs formed a stable multibody and stuck to the expected target site."

I don't understand what "many EcoCascade RNPs" means. Does it mean that many EcoCascades are bound to the target site of one dsDNA? If so, the authors should indicate them on the figures or the movies.

15. page 13, lines 298-300, "Notably, we observed a typical DNA bend at the EcoCascade-RNP binding site for stable R-loop formation, as previously indicated by cryo-EM and smFRET studies."

In the video1, dissociation of EcoCascade-RNPs from the dsDNA can also be seen. The authors are requested to analyze the binding times of EcoCascade-RNPs on dsDNA and have a quantitative discussion.

16. page 14, lines 308-309, “Notably, the shape of DNA bending was similar to that of artificially nicked dsDNA using Nb.BsrDI nicking endonucleases (Extended data Fig. 13b).”

This result needs to be discussed quantitatively by creating a histogram showing the distribution of bending angles of EcoCans3 bound dsDNA and SSB-like DNA.

17. page 14, lines 312-313, “In contrast, in ATP-containing reaction buffer, we detected many DNA fragments of 219-bp or 423-bp after injection of EcoCas3 proteins.”

The AFM images to show this result are not demonstrated. The relevant AFM images and quantitative length analysis of DNA fragments should be shown.

18. The playback of video5 is too fast and it is very difficult to understand what the authors are trying to show. The dynamic phenomena in the video should be displayed clearly, such as slowing down the playback of important scenes or adding illustrations along with DNA in the video.

19. To be honest, the image quality of Fig4d is so poor that I'm not sure if what the authors claim is plausible. Can't the authors make the figure more convincing with proper image processing to enhance the contrast?

Point-by-point response to the reviewers' comments

Reviewer #1 (Remarks to the Author):

In this manuscript, Yoshimi et al. describe a new activity of Cas3, the endonuclease effector of the type I-E CRISPR-Cas system. Type I-E is among the best studied CRISPR-Cas system due to its presence in *E. coli*. Although extensive mechanistic and structural studies have been performed for this system, several questions remain. In particular, this study addresses questions related to how Cas3 is able to degrade both strands of the DNA target, when only one strand (the non-target strand) is known to be threaded in the nuclease active site. The major new finding of this paper is that Cas3 has trans collateral cleavage activity when activated by Cascade, similar to Cas12a collateral cleavage upon target activation. The authors demonstrate this activity very convincingly, and also perform interesting dsDNA cleavage assays that show how the target strand gets cleaved by Cas3 in trans. The authors also perform high-speed AFM experiments that allow visualization of the interference process. Although the videos of the AFM experiments are difficult to follow for the untrained eye, the authors provide snapshots in the static figures that help the reader follow the process.

The AFM experiments are an interesting inclusion, but the most important finding is the trans cleavage by Cas3. This cleavage activity adds to a growing list of activated nickases that have been reported for Cas proteins. Given the recent development of CRISPR-Cas3 genome editing as well as Cas12a, Cas13 and Csm6-based diagnostics, it is likely that this finding will generate interest not only among the CRISPR community, but also in the applications fields. I have a few concerns about the way data is presented (listed below), but otherwise I am supportive of publication.

We are very grateful to this reviewer for their positive comments on our manuscript. Below, we reply point-by-point to the concerns raised by the reviewer.

1. Many of the results are recapitulations of well-known Cascade-Cas3 activities. The authors could consider shortening the Results section

significantly by moving description of Cas3 purification to the Methods and reducing discussion of the effects of mismatches on target binding. The data shown in Figs. 1b-c are controls showing that the system the authors have reconstituted works similar to prior studies, and could be described more briefly.

This comment and the comments below have enabled us to much improve our manuscript and to make it more clearly understandable for readers. In accordance with this comment, we reduced the description of Cas3 purification, pages 5–6, and transferred Figs. 1b,c to Extended data Figs 6a,b to shorten the target binding description on page 8, lines 15–21.

2. The collateral cleavage with ssDNA targets show interesting differences between Cas3 and Cas12a. A likely explanation for this difference is due to inefficient recruitment of Cas3 due to Cas8 conformation. It is known that Cas8 adopts a different conformation when bound to ssDNA (see refs 24 and 59), which is thought to reduce Cas3 recruitment. This could be added to the Results or Discussion section. On the other hand, it is unclear why the PAM sequence makes a difference in this case. It has previously been thought that PAM must be double-stranded for its recognition, consistent with its recognition in crystal and cryo-EM structures. It might be interesting to do an experiment with the same ssDNA where the PAM region is double stranded but the rest of the protospacer is single stranded (i.e. using a partially double-stranded DNA in which only the PAM and upstream region are double-stranded). With a double-stranded PAM allowing proper Cas8 conformation, Cas3 recruitment may be enabled and the partial ssDNA may fully activated trans cleavage.

We totally agree with the first comment regarding the conformational change of Cas8 when bound to ssDNA. Accordingly, we added a discussion including two references on page 17, lines 1–4. We also appreciate the second comment requesting an additional experiment using the assay for ssDNA-activated collateral cleavage. As predicted by the reviewer, we found interesting results; the same ssDNA with a double-stranded PAM showed higher activity compared with that of either ssDNA- or even dsDNA-activated collateral cleavage (new Fig. 1f). We discussed this interesting phenomenon on page 18, lines 3–6.

3. It is surprising that LbaCas12a is so permissive of PAM sequences for

collateral cleavage. This is, to my knowledge, inconsistent with prior publications (e.g. Chen et al. 2018), which show the expected PAM-dependence for Cas12a dsDNA targeting. Do the authors have an explanation for this?

In our experiments, LbaCas12a surprisingly showed that PAM sequences were permissive for collateral cleavage. Although it is difficult to explain, we discussed the potential reason on page 17, lines 22–25.

4. In the cleavage experiments with the NTS and TS labeled, it would be helpful to not only label the location of the PAM, but also the location of the full length DNA and the location of the protospacer in each chromatogram. The protospacer could be indicated by shading or a box. It is very difficult to read the numbers on the x-axis of these chromatograms, so a sense of scale is missing from the figure.

We appreciate this comment. Accordingly, we added an explanation of the chromatograph to the figure legend: 'The x-axis represents the DNA fragment size in base pairs and the y-axis represents the fluorescence units'. We also added the PAM and spacer sites to the figures. We hope that the meaning is now easily understandable.

5. Many figures have very small font sizes, making it hard to read. Please adjust all font sizes to at least 6 point font.

We apologize for the small font sizes. We changed all fonts to 6 point or larger in all figures.

6. On 389-390 what is meant by "which carry small mutations"? Do you mean that these create DSBs rather than long range cleavage events? Please clarify, or potentially remove this sentence, which seems speculative.

In accordance with the comment, we deleted this sentence on page 17, lines 12–15.

7. There are several figures where Cascade is spelled Casscade (Fig 1b, 1c,

Extended data figure 2c)

Thank you for alerting us to this error. All figures have been corrected.

8. In general, the references could be updated a bit to better reflect the literature. I urge the authors to go through the introduction and discussion and make sure the correct and comprehensive references are cited to support individual statements. A few specific examples are listed below.

a. On line 59-60, the authors cite a number of references, including 2 papers reporting the first crystal structures of Cascade. Another paper reporting the same structure (PMID 25118175) was published at the same time but was omitted from these references. Please add this as well.

We appreciate the respectful comments regarding to the lack of references throughout the manuscript. Accordingly, we have added the reference (PMID 25118175) on page 4, line 20.

b. On line 337, these two papers used single molecule fluorescence assays (DNA curtains assays) but not smFRET. Similarly, on line 346, only one of these papers (57) was an smFRET study. Please update these sentences.

According to the comment, we corrected the sentences on page 15, line 18 and on page 16, lines 1–2, respectively.

c. The sentence at lines 357-359 seems redundant with other parts of the paragraph. This could be mentioned when priming is first mentioned at the beginning of the paragraph. It would also be appropriate to cite the first papers to demonstrate priming in vivo (Datsenko et al, 2012 and Swarts et al, 2012)

According to the comment, we deleted the redundant sentences on page 16, lines 1–4 and added the two references that first demonstrated priming in vivo. Again, we appreciate these comments.

d. Line 372: not all of these studies are cryo-EM studies, some of them report crystal structures.

e. Line 381: Only ref 66 reported a crystal structure of Cas3, ref 30 reported a

cryo-EM structure of Cascade-Cas3 complex.

According to the comment, we changed the sentences on page 16, line 25 and on page 17, line 4.

Reviewer #2 (Remarks to the Author):

1. Conflicts of interest: Several of the authors are from a company. The interest of the company in this work should be explained so that the readers can make their own judgements about potential biases.

In accordance with the comment, we state the competing interests of authors on page 20, lines 1–4.

2. General HSAFM methods description: The authors cite reference 55 for some of the key methodological details and it would be much more helpful for them to include the relevant information in the manuscript or supplementary info, versus expecting the reviewer to search through older references.

We appreciate the reviewer's suggestion. Accordingly, we added the relevant methodological details to pages 28–31. We also added several references for hs-AFM methodology on page 29, line 22 and on page 30, line 2.

Details of the scanning parameters used in this study need to be added - both for the APTES surface experiments and the two different lipid bilayer experiments. Specifically, what was the probe linear velocity across the surface? What was the scan dimension and frequency in the fast scan and slow scan axes? Engagement of the probe with the surface before scanning is a very important part of the procedure. This needs to be described. What was the velocity profile as the probe was moved vertically to engage the surface? What force conditions or other parameters were used to indicate a successful engage condition? This is important to know, because in some cases, these parameters can be different from the steady-state feedback set point conditions used during imaging.

We apologize for the lack of important information. In accordance with the comment, we added details of the scanning parameters to the legend of each

HS-AFM image and to the method section on page 28, line 15. We followed the reference protocol {Uchihashi, Nature Protocol 2012} for engaging the probe with the surface before scanning in the AFM experiment and for adjusting the feedback parameters.

The authors need to include some comments about the difficulty and reproducibility of the imaging. How many experiments are required to get a successful result? How many different regions are interrogated on the surface during a single experiment? How many repeats were conducted? Was the DNA evenly distributed on the surface or did the operator need to hunt for a suitable region to image before proceeding? Is there a time after adding the protein complexes that the system is no longer viable or good data can no longer be recorded? i.e. how fast does the experiment need to be conducted?

We appreciate your valuable comments and understanding of the realities of AFM experiments. We had no trouble in obtaining the shape features of the molecule at high resolution on the APTES mica and the experiment was completed quickly. However, we had to optimize many substrate and solution conditions before we could establish the experimental conditions to observe Cascade binding to dsDNA and the phenomenon of dsDNA reeling and DSB induced by the addition of Cas3 and ATP. Finally, having established the experimental conditions, at least three independent experiments were performed and the same results were obtained. This ensured that the experiments were reproducible, which we have described in the Methods. The coarse observation range of our scanner is $2 \times 3 \mu\text{m}^2$. We adjusted the concentration of DNA and DNA-Cascade complexes so that they were encountered at a rate of at least one per $500 \times 500 \text{ nm}^2$ area. Thus, the operator could easily find the target molecules.

While the lipid membrane system has the advantage of tracking the reactions of biomolecules, it also has the disadvantage of the probe being easily fouled. As a consequence, the image quality often decreased drastically. In such cases, it was necessary to wait until the tip of the probe was cleaned or to replace the cantilever. This required some patience because no good data could be recorded during this time. If the cantilever or the substrate had to be replaced, it took about 30 minutes of setup. The addition of 10% glycerol had a positive effect in keeping the probe clean.

As for how fast we had to conduct the experiment, we made sure that the protein concentration was such that the reaction would occur gradually, so that we did not have to rush the experiment. We encountered more already-reacted molecules as time went by, but we could find unreacted molecules and reactions to these molecules for about 15 minutes after the addition of proteins. We added a summary of this information on page 29.

Many of the sample prep details that would be required for other scientists to reproduce these experiments are missing. This is an important deficiency because AFM experiments often hinge on successful sample prep as much as, or more than, on instrument operation. For example, a lipid supported bilayer was used as a substrate for some of the observations. The procedures for preparation of the bilayer and application of the bilayer to the mica surface need to be described in sufficient detail such that they could be reproduced by others. The solution conditions for the DNA amplicons applied to the bilayer also needs to be described, as does the volume of liquid applied and the surface area of the substrate that is covered. Is the DNA solution removed or otherwise discarded before the wash step? What is the estimated final surface coverage of the DNA? i.e. how many molecules are retained on the surface per square micron. This is relevant because the molar ratio of protein complex to DNA strand needs to be estimated so that kinetics can be inferred, as well as for practical purposes of reproducing the experiment.

We apologize for the lack of description of many experimental procedures related to sample preparation and lipid membrane preparation in the manuscript. As pointed out, the preparation of liposomes and their application to the mica surface requires some manual effort, and is the same method described previously in our *Nature Communications* article. However, we have included it to for the reader's convenience. We have also cited references in *Nature Protocols* and *Methods in Enzymology* that explain the mica-SLB method in detail in the methods on page 30.

The solution for placing the DNA on the lipid membrane was Milli-Q water. The volume loaded was 2 μ l and unattached DNA molecules were washed away. The surface of the substrate was a 1.5 mm diameter mica disk (Furuuchi Chemical). Concerning the DNA coverage, DNA molecules were

found at least one DNA per 500 x 500 nm² area. We added these important information to our method on page 31.

For the lipid bilayer experiments, is the observation buffer the same as described in the preceding paragraph, for the APTES surfaces? Is the mica substrate fixed to the scan stage during this procedure, or is it fixed after DNA application and washing? Details of the liquid cell design and geometry should be included, as well as a schematic of how the cell and the HSAFM scan stage are assembled together. If already described in a previous publication, a cursory description and reference should be included in this manuscript. The EcoCascade was added as some point after the sample was in the instrument. What were the solution conditions for the EcoCascade sample added? The room was heated to 30C -- was the fluid cell and the sample liquid equilibrated at the same temperature? If so, how long was the EcoCascade sample maintained at 30C after it was taken from cold storage and introduced to the cell?

We apologize for the unclear description of the method. The same observation solution was used for each experiment. Details of the liquid cell design and geometry can be found in the references cited (*Nature Protocols* and in Nonaka et al.). In the experiments where EcoCascade was added to the solution cell, EcoCascade was diluted in the observation solution. We have added this information to the method. The liquid cell and the sample solution had a volume of 60 μ L and had been placed in a room at 30°C for more than a few minutes, so they had equilibrated sufficiently.

The high concentration of stock EcoCascade was diluted with the observation solution that had been placed in the room at 30°C, and 6 μ L was taken with a pipette that had been placed in the room at 30°C. The added EcoCascade solution was also sufficiently equilibrated at 30°C. The EcoCas3 and ATP solutions were prepared in the same way. This information was added to the method on page 30, lines 1–19.

Why was a different bilayer preparation used for observation of EcoCascade binding versus Cas3 reeling and double-strand break behavior? In the reeling and ds break experiments, the sample was maintained at a temperature of 37C -- why was the temperature different from the earlier experiments (30C). Is this

change inconsequential, or relevant to the procedure?

We appreciate your valuable comment. As DPTAP has a positively charged head group, addition of DPTAP facilitated immobilization of negatively charged DNA on mica-SLB. As biotin-cap-DPPE has a bulky head group, addition of biotin-cap-DPPE induced sub-nanometer roughness on mica-SLB. This increased the mobility of dsDNA on the surface and sometimes allowed us to observe the binding events of proteins that surround dsDNA as previously seen in the observation of CRISPR-Cas9. This information was added to the methods on page 31, lines 9–15.

We could observe the target binding of EcoCascade on the mica-SLB surface using the lipid composition of 90:5:5 (w/w) DPPC:DPTAP:biotin-cap-DPPE. However, this substrate did not allow us to observe the reeling and reel-out events after addition of Cas3 and ATP, although we could observe the reaction products made by the DSB events and rarely observe DSB events, in which the reeling events were already over before imaging on this substrate. This indicates that this substrate does not completely inhibit the reaction by Cas3, while significantly reducing the efficiency of the reaction.

In contrast, we succeeded in observing real-time reeling, reel-out and DSB events upon addition of Cas3 and ATP on the mica-SLB surface using the lipid composition of 80:10:10 (w/w) DPPC:DPTAP:biotin-cap-DPPE. As biotin-cap-DPPE has a bulky head group, addition of biotin-cap-DPPE induced sub-nanometer roughness on the mica-SLB and increased the mobility of DNA on the surface. This might allow Cas3 to function on the surface. We believe that the latter substrate would also be useful for observing the target search by EcoCascade although we have not tried it. We performed the DSB experiment at 37°C because this is a suitable temperature for the DSB reaction. This information was added to the methods on page 30, lines 20–25.

3. HSAFM data. In general, the quality of the HSAFM images is poor, to the point that many of the claims I cannot evaluate. Furthermore, in these kinds of observational experiments, many events and the associated statistics need to be presented before drawing conclusions. Drawing conclusions from observation of one or two molecules when there are likely millions on the surface in a single experiment and multiple repeats could be conducted, is not

methodologically sound, in my opinion. For these reasons, I find that this group of experiments detracts from, rather than adds to, the quality of the manuscript.

We appreciate your important suggestion. We performed additional experiments for the statistical evaluation of these HS-AFM observations. These experiments and results were added to page 13, line 10 to page 14, line 23 and Extended data Figs. 15–20.

Figure 4a. This is a cartoon, not an observation. The manuscript (line 291..) describes it as an observation, which is confusing.

According to the comment, we changed the sentence on page 13, line 6.

Figure 4b and video 1 and video 2. Video 1 and the frames abstracted from it are fairly clear. Video 2 claims to show a bound complex and R-loop architecture. However, video 2 is too blurry for this reviewer to interpret. It appears that the protein complex binds to a strand but does not move.

As you mentioned, Video 2 was difficult to interpret for readers. To facilitate understanding, we have filtered the video to make it clearer. We have also added another clearer video.

Importantly, the manuscript indicates (line 297..) that many EcoCascade RNPs formed stable complexes and bound to the target site. However, it is not clear what data (which videos and images) are used to make this conclusion. Also claimed is that the complexes bound tightly to the target DNAs without dissociation. In video 1, there appears to be a complex of proteins moving along the DNA, while also occasionally desorbing and re-adsorbing. It is not clear if this is a complex or a partial complex.

We appreciate your important comments. To show that EcoCascade binds specifically and strongly to the target locus, we anchored a premix of EcoCascade and dsDNA on APTES mica to determine where Cascade binds on the full length dsDNA (N = 47). We confirmed that Cascade binds specifically to about 67% of the total dsDNA length, which is consistent with the location of the target site (Fig. 4a). These Cascades bound to the same site for more than

10 s in all cases unless a strong tapping force was applied. In contrast, the duration of the non-target Cascade was mostly measured to be less than 1 s, suggesting that it binds intensely only at the target site. In addition, the contour length of Cascade-free DNA was observed to be 232 ± 10 nm, whereas the contour length of Cascade-bound DNA was observed to be 216 ± 17 nm, about 16 nm shorter. This suggests that the DNA was lifted off the substrate by the incorporation of Cascade into the DNA and is consistent with the structural observations from electron microscopy.

In addition, because we purified the single-peak protein by gel filtration, we believe that there were few incomplete complexes. Cascade is shown in Video 1 sliding through the DNA and then binding to the target site. This was not an incomplete complex reaction because such reactions are often reproduced in free DNA regions. Interestingly, the binding time of the non-target Cascade was fitted with a double exponential curve (Extended data Fig. 16). This suggests that Cascade has two binding modes. Such sliding has been seen in previous papers and we believe it is reproducible (Cell Rep 2017 doi: 10.1016/j.celrep.2017.11.110.). We have updated the text to include these additional data and discussion in Fig. 4b, Video 1 and 2.

Figure 4c and video 4. This video and associated frames appear to show two protein complexes bound to two different DNA strands at roughly the same relative position from the end from each strand. The relative position corresponds to where the expected target site would be, though there is nothing to ensure that this is the case for these molecules because the two ends are not individually distinguishable. How many different times was this behavior observed? Can we be sure that this is not a random observation? The video/figure also claims to show the formation of a nick site, but this entirely based on a kink structure in the DNA backbone, not actual observation of the nick itself. Since there are many possible causes of a kink, including a nick, this observation is not particularly compelling without some other sort of identification, such as enzymatic labeling of the actual nick.

As suggested by the reviewer, the ends of DNA are indistinguishable in these experiments. However, Cascade only binds intensely at the target site, which was demonstrated by the results of the binding position of Cascade to DNA (Extended

data Fig. 15) and the lifetime analysis (Extended data Fig. 16). Thus, we only focused on Cascade binding stably for more than 10 seconds for further analyses.

We appreciate your suggestion about the nick site. To identify the nick site by hs-AFM directly rather than the kink structure, we measured the height of DNA at the target site because nicking lowers the height of DNA structures. After the reaction of Cas3 and Cascade complex with dsDNAs but without ATP, EcoCascade-EcoCas3 proteins were removed from dsDNA by a strong tapping force via the AFM probe. Observation of the complex binding sites showed a significant decrease in the height of DNA at about 60% of sites (55/89 loci, 1.34 ± 0.24 nm vs 1.96 ± 0.16 nm) (Extended data Fig. 18a, b, Video 3 and 4). In contrast, no nick structure was observed after mechanically removing EcoCascade from DNA without EcoCas3 protein (Extended data Fig. 18c, d). Therefore, we could provide visual evidence that Cas3 can induce nicking at the target site. Furthermore, we have confirmed in Fig. 3 that the nick occurs by the reaction of Cas3 in the absence of ATP and in the dead helicase Cas3 by fragment analysis. The observation of nicking by hs-AFM in Fig. 4c supports these results. These data and explanations were added on page 14, line 4–13.

Figure 4d and video 6. The image contrast and noise character of these images are too poor for this reviewer to interpret them. Since discussion of 'reeling' observation is based on this data, I cannot evaluate the claims in the manuscript.

As you mentioned, Figure 4d and Video 6 were difficult to interpret for readers. To facilitate understanding, we have made some changes, such as filtering the AFM images to make them easier to read and highlighting important frames.

Reviewer #3 (Remarks to the Author):

Yoshimi et al. have elucidated the molecular mechanism of CRISPR interference by the CRISPR-Cas3 system by employing a variety of biochemical and biophysical techniques. The most important aspect of this work is the finding that the binding of type I CRISPR Cas effectors to target dsDNA has two modes depending on the binding strength, switching between cleavage of the collateral ssDNA and the target dsDNA and proposing a mechanistic model for type I

CRISPR priming and interference against a foreign DNA. Direct observations of dynamics by hs-AFM are really interesting because of the first direct observation of the sequence of events from DNA binding to dsDNA cleavage in the Cas3 system at the single molecule level, but it does not seem to be directly related to the main claims of this manuscript. In fact, it is a bit strange to discuss hs-AFM data at the beginning of discussion when it is not even mentioned in the conclusion. Rather, it would be more effective to demonstrate the use of hs-AFM by imaging the strength of the binding between EcoCascade and the target DNA and the fluctuation of the binding, as in the experiment shown in Fig.3d. Since the conclusions of the manuscript itself are intriguing and provide mechanistic insights of CRISPR interference, it is worthy of eventual publication in Nature Communications. However, the technical terms that are key to understanding the overall argument are not well defined and the explanations are very difficult to understand. In particular, the definition of trans- and cis-cleavage was not clear to the reviewer who is not an expert of CRISPR systems, even though the important point of this paper cannot be understood without knowing it. It should be explained first so that non-specialist readers can understand it even if it is obvious to experts. Also, data is not properly selected, making the manuscript redundant and difficult to read. There is a lack of quantitative analysis regarding HS-AFM data, which is too intuitive. That's important, but the discussion should be based on quantitative analysis whenever possible. Below are some questions and comments that should be considered for revising the manuscript.

1. First of all, please explain the meaning of trans- and cis-cleavage with an illustration in the introduction. Though it may be obvious to experts, it is quite hard to read the manuscript without understanding the meaning of terms.

We appreciate the comments of the reviewer, which have enabled us to greatly improve our manuscript. In accordance with this first comment, we have explained the meaning of trans- and cis-cleavage in the introduction, page 4, lines 18–20, and added a description to Figure 1a,c to facilitate understanding.

2. In Fig. 1d, the authors investigated the effect of divalent ions on non-specific cleavage of ssDNA, but I don't understand what the authors are trying to claim with this result. Is there any effect of divalent ions on ssDNA cleavage in the presence of the target dsDNA?

There are several papers reporting that Cas3 protein exhibits indiscriminate, divalent cation-dependent ssDNase activity in the absence of Cascade (Sinkunas 2011, Beloglazova 2011, Mulepati 2011), as described on page 6, lines 9–11. This divalent cation-dependent activity may affect not only target dsDNA-activated ssDNase cleavage, but also target dsDNA degradation. To rule out this activity, we showed that EcoCas3 when used alone did not show any indiscriminate cleavage activity with Mg^{2+} or Ca^{2+} .

3. page 8, lines 161-163, “In ATP-free reaction buffer (-), the collateral activity of the EcoCas3 protein was at the same level as that of wild-type EcoCas3 and the dhCas3 mutant in ATP (+) buffer (Fig. 1f).”

The authors concluded that the collateral ssDNA cleavage activity of EcoCas3 in ATP-free buffer is the same level as that in the presence of ATP. However, it is difficult to conclude that to me that they are at the same level, simply looking at this graph. The cleavage activity appears to be higher in the ATP-free condition. The authors should clearly explain the basis for concluding that the activity is at the same level. If the activity is increased in ATP-free, the possible cause should be mentioned.

We agree that it is difficult to conclude that the collateral ssDNA cleavage activity in ATP-free buffer is at the same level as that in the presence of ATP by simply comparing the graph values. It is also difficult to explain the reason for the increased activity in ATP-free conditions. Therefore, we simply described the results on page 7, lines 15–21.

4. page 8, line 163-165, “Together, these results indicate that the nuclease and helicase activities of EcoCas3 are required for target DNA degradation, but only the nuclease activity is required for collateral cleavage.”

For dnCas3, does it have the helicase activity of dsDNA without the cleavage activity? I mean, does the unwinding of dsDNA occur by dnCas3?

The dnCas3 should have helicase activity for dsDNA, but we have no evidence for this. We have changed the description for the helicase and nuclease activity on page 7, lines 15–21.

5. page 8, lines 177-185, “In contrast, LBaCas12a...”

I don't understand why the authors need to show the Cas12a results in this manuscript even though the results are the same as previous studies. Shouldn't they just focus on the Cas3 results?

Several papers have already reported collateral cleavage for Cas12a but not for Cas9. Therefore, we have used it as a control not only for PAM specificity but also for crRNA-complementary ssDNA-activated collateral cleavage in the subsequent experiments. Importantly, Swart et al reported mechanistic insights into the cis- and trans-acting DNase activities of Cas12a. We consider that comparison of PAM specificity and collateral cleavage activity between Cas3 and Cas12a is important and we think it is appropriate to include the Cas12a results.

6. pages 8-9, “PAM recognition is a prerequisite for collateral ssDNA cleavage by Cas3 but not Cas12a.”

The experimental results shown in Figs. 2a-e are based on the NTS PAM sequence, but Figs. 2c and d are explained by the TS sequence, which is very complicated. Fig. 2a-e should also be shown with the PAM sequence of the TS.

We appreciate this comment. Accordingly, we have shown the PAM sequence of the NTS and the TS in Fig. 2a–e.

7. page 9, lines 190-192, “..., except for when the third nucleotide of the PAM was C, such as TAC, AGC, GTC, GAC, and GGC,...”

What are the implications of these results? Are there any possible explanations for the results in the model description?

In accordance with this comment, we have added ‘This means the third nucleotide of the TS PAM is important for the collateral cleavage’, page 8, lines 24–25. We have also added discussion to, page 18, lines 1–9: ‘In the CRISPR-Cas3 system, EcoCas3 recruitment and binding to EcoCas8 depend on TS-PAM recognition. EcoCas8 binds to the third position of the TS-PAM and unwinds through recognition of the NTS-PAM (Hayes, 2016), which may increase PAM specificity in collateral ssDNA cleavage (Fig. 2)’.

8. page 9, lines 196-198, “This is not consistent with a previous report that showed dsDNA with an unpaired PAM did not activate EcoCas3 to degrade target DNA substrates.

Fig. 2f shows the result of collateral ssDNA cleavage activity, which is not different from the degradation of the target DNA. I don't understand what the authors say “not consistent”.

In accordance with the reviewer's comment, we changed the sentence to “This is in contrast to previous reports of dsDNA with an unpaired PAM not showing any activity for target DNA degradation {Westra 2013, Hochstrasser 2014}.”, on page 9, line 5–6.

9. I have no idea how to look at the data shown in Fig.3a-c, and there is no explanation even in the method section. There is no description of the meaning of the horizontal and vertical axes of the graph, and even the meaning of the colored dashed lines in the figures. I understand what the authors mention from the context, but it should be explained properly so that everyone can understand.

In accordance with this comment, we added an explanation of the chromatograph to the figure legends: ‘The x-axis represents the DNA fragment size in base pairs, and the y-axis represents the fluorescence units.’ We also added the PAM and spacer sites in the figures. We hope that this clarifies the meaning for readers.

10. page 10, lines 227-229, “Notably, the dhCas3 mutant cleaved the NTS in cis, but not the TS in trans (Fig. 3b), which was not consistent with the assay's collateral cleavage results where the dhCas3 mutant cleaved non-specific ssDNA in trans (Fig. 1f).”

I don't understand the meaning of "the NTS in cis" and "the TS in trans" in this sentence. Why not simply the NTS and the TS? Same as the very first question, it should be explained clearly since it seems the key point of this manuscript.

In accordance with this comment, we provide a simple explanation of the NTS and the TS, page 10, line 16. We also simplified the following sentences in

terms of cis and trans. We agree that this is the key point of this manuscript; therefore, we explained this clearly in the results and the conclusion, page 10, lines 17–22.

11. page 11, lines 238-242, “We also observed that progressive cis and trans cleavages showed similar patterns in the repetitive experiments and the short and long incubation experiments, depending on the target DNA sequence (Fig. 3 and Extended data Fig. 10a). The sizes of many cleaved fragments were between 30–60 bps, which may be used for CRISPR adaptations as previously reported (Fig. 3 and Extended data Fig. 10b).”

I have no idea what Fig.3 in these sentences refers to or what it is talking about. Is the figure number correct?

We would like to mention that we observed similar cleavage patterns of DNA fragmentation when CRISPR-Cas3 acted on the same target sites, such as *hEMX1* and *mTyr*, in the repetitive experiments. We corrected the figure numbers to Extended data Fig. 12a and 12b.

12. page 11, lines 251-252, “ In the LbaCas12a system, 1–3 mismatches in the seed region also did not affect collateral cleavage activity (Extended data Fig. 11c), consistent with previous reports”

Is it really necessary to show the data even though it is only to confirm the previous results?

As suggested, we have confirmed the results in previous reports here. In this manuscript, we discovered the collateral cleavage activity of the CRISPR-Cas3 system, its PAM specificity, and the tolerance of mismatches in the seed region. We compared them with those of the CRISPR-Cas12a system because they have been previously reported. We do not think we should delete this comparison data derived under our experimental conditions, even if the results are the same as those gained previously.

13. page. 13, lines 291-294, “First, we visualized the binding of Cascade/crRNA to a target DNA, a 645-bp dsDNA containing a target spacer site flanked by a PAM (AAG) at 219-bp and 423-bp from the ends of the DNA fragment (Fig. 4a). We then adsorbed the mixture of donor DNAs and EcoCascade RNPs onto a 3-

aminopropyl-triethoxy silane-mica surface (APTES-mica).”

The two sentences should explain the conditions of different experiments. This way of writing misleads readers into thinking that it is a continuous experiment. The second condition should be mentioned when showing the corresponding data.

We apologize for our uninformative description. We only performed one experiment for visualizing the binding of Cascade/crRNA to a target DNA. We made it clear that we conducted the experiment on page 13, lines 8–19.

14. page 13, lines 297-298, “We also found that many EcoCascade RNPs formed a stable multibody and stuck to the expected target site.”

I don't understand what "many EcoCascade RNPs" means. Does it mean that many EcoCascades are bound to the target site of one dsDNA? If so, the authors should indicate them on the figures or the movies.

We apologize for our description. As suggested, we made the sentences clear on page 13, lines 8–19.

15. page 13, lines 298-300, “Notably, we observed a typical DNA bend at the EcoCascade-RNP binding site for stable R-loop formation, as previously indicated by cryo-EM and smFRET studies.”

In the video, dissociation of EcoCascade-RNPs from the dsDNA can also be seen. The authors are requested to analyze the binding times of EcoCascade-RNPs on dsDNA and have a quantitative discussion.

We appreciate for these important comments. To show that EcoCascade binds specifically and strongly to the target locus, we anchored a premix of EcoCascade and dsDNA on APTES mica to determine where Cascade binds on the full length dsDNA (N = 47). We confirmed that Cascade binds specifically to about 67% of the total dsDNA length, which is consistent with the location of the target site (Extended data Fig. 15). These Cascades bound to the same site for more than 10 s in all cases unless a strong tapping force was applied. The duration of the non-target Cascade was mostly measured to be less than 1 s, suggesting that the Cascade binds intensely only at the target site. We have

updated the text to include these additional data and discussion in Extended data Fig. 16.

16. page 14, lines 308-309, “Notably, the shape of DNA bending was similar to that of artificially nicked dsDNA using Nb.BsrDI nicking endonucleases (Extended data Fig. 13b).”

This result needs to be discussed quantitatively by creating a histogram showing the distribution of bending angles of EcoCas3 bound dsDNA and SSB-like DNA.

We appreciate your important comments. We considered the angle analysis as you mentioned but found it difficult to evaluate the nick because the angle can change by Cascade binding only, as shown in Extended data Fig. 15. Therefore, rather than determining the shape of DNA bending, we measured the height of DNA at the target site because nicking lowers the height of DNA structures, which can identify the nick site directly and quantitatively. After the reaction of Cas3 and Cascade complex with dsDNAs but without ATP, EcoCascade-EcoCas3 proteins were removed from dsDNA by the strong tapping force of the AFM probe. Observation of the complex binding sites showed a significant decrease in the height of DNA at about 60% of sites (55/89 loci, 1.34 ± 0.24 nm vs 1.96 ± 0.16 nm) (Extended data Fig. 18a, b). In contrast, no nick structure was observed after mechanically removing EcoCascade from DNA without EcoCas3 protein (Extended data Fig. 18c, d). Therefore, we provide visual evidence that Cas3 can induce nicking at the target site. These data and explanations were added on page 14, lines 4–13.

17. page 14, lines 312-313, “In contrast, in ATP-containing reaction buffer, we detected many DNA fragments of 219-bp or 423-bp after injection of EcoCas3 proteins.”

The AFM images to show this result are not demonstrated. The relevant AFM images and quantitative length analysis of DNA fragments should be shown.

As commented, we should have shown the results of the quantitative analysis of DNA fragments. To perform a quantitative analysis of the length of DNA fragments, we added Cas3 and ATP to the Cascade-DNA premix on the substrate and analyzed the length of short DNAs observed after the reaction.

We observed 140 cases and, interestingly, we obtained many DNA fragments that were shorter than expected (Extended data Fig. 19). This result suggests that DSBs are introduced not only at the target site where Cascade binds, but also at unspecified sites upstream of the target site. These results are consistent with the finding of DNA cleavage upstream of the PAM and with the results of fragment analysis shown in Fig. 3. We added these results and discussion to page 14, lines 14–23.

18. The playback of video5 is too fast and it is very difficult to understand what the authors are trying to show. The dynamic phenomena in the video should be displayed clearly, such as slowing down the playback of important scenes or adding illustrations along with DNA in the video.

As commented, our videos were difficult to interpret for readers. To facilitate understanding, we made some changes, such as filtering the AFM images to make them easier to view, and we highlighted important frames.

19. To be honest, the image quality of Fig4d is so poor that I'm not sure if what the authors claim is plausible. Can't the authors make the figure more convincing with proper image processing to enhance the contrast?

We changed Fig. 4d to facilitate understanding as suggested.

REVIEWERS' COMMENTS

Reviewer #1 (Remarks to the Author):

The authors have mostly addressed my concerns. I do have a few more issues with the PAM requirement sections of the Results and Discussion, listed below.

1. There are some lines where the authors mention the sequence of the PAM on the target strand, rather than the non-target strand (which is standard in the field). When mentioning the target strand sequence, they list the sequence in the 3'-5' direction, i.e. TTC or AAAC. It would help to indicate that these are being described in this direction. I would suggest that the authors indicate 5' and 3' end of all PAM sequences that are listed in the text and be clear whether they are talking about the TS or NTS sequence.

2. I am not satisfied with the explanation of why LbCas12a did not show any PAM dependence for trans ssDNA cleavage. This trans cleavage requires target binding and cleavage prior to activation of LbCas12a. Binding of dsDNA targets requires PAM recognition by LbCas12a, as it does by Cascade and Cas9 as well. Therefore, if a dsDNA is being used as the activator, it would stand to reason that some level of PAM specificity would be observed for the trans cleavage activity, similar to what would be observed for targeted dsDNA cleavage activity. One potential explanation for why this was not observed is that the dsDNA targets are not fully double stranded, and some ssDNA contamination is sufficient to activate LbCas12a. Such an issue would not arise for Cascade, given that ssDNA does not activate Cascade-Cas3 trans cleavage as well as a dsDNA target or a target containing a double-stranded PAM.

Reviewer #2 (Remarks to the Author):

I support publications of this version.

The authors have done a very good job of answering my questions and the community will appreciate the detail.

I do think the editor should ask for a bit more disclosure about the relationship between the company and this work, so that readers can make their own judgements about any potential bias.

Reviewer #3 (Remarks to the Author):

The authors' appropriate revisions have satisfied all my questions, especially about terminology, the results of biochemical analysis, and their significance. The main HS-AFM image itself has not been updated, and its quality is still not entirely satisfactory, but the addition of the illustration helped me understand what the author was trying to state from the AFM images. I also commend the authors for the great effort they put into the quantitative analysis of the AFM data, which makes the HS-AFM data more convincing. On the other hand, what most of the results of the analysis of the HS-AFM data are included in the extended material raises me questions about the organization of the paper considering the description quantity in the main text. If it is possible due to the character limit of the journal, it would be better to move some HS-AFM data and analysis to the main text.

Point-by-point response to the reviewers' comments

We appreciate all the comments the reviewers raised in the revised manuscript. Those comments have enabled us to improve the final version of our manuscript. We made point-by-point responses to the reviewers' comments below.

Reviewer #1:

The authors have mostly addressed my concerns. I do have a few more issues with the PAM requirement sections of the Results and Discussion, listed below.

1. There are some lines where the authors mention the sequence of the PAM on the target strand, rather than the non-target strand (which is standard in the field). When mentioning the target strand sequence, they list the sequence in the 3'-5' direction, i.e. TTC or AAAC. It would help to indicate that these are being described in this direction. I would suggest that the authors indicate 5' and 3' end of all PAM sequences that are listed in the text and be clear whether they are talking about the TS or NTS sequence.

In accordance with this comment, we indicated 5' and 3' end of all PAM sequences that are listed in the manuscript as well as the TS or NTS in the figures.

2. I am not satisfied with the explanation of why LbCas12a did not show any PAM dependence for trans ssDNA cleavage. This trans cleavage requires target binding and cleavage prior to activation of LbCas12a. Binding of dsDNA targets requires PAM recognition by LbCas12a, as it does by Cascade and Cas9 as well. Therefore, if a dsDNA is being used as the activator, it would stand to reason that some level of PAM specificity would be observed for the trans cleavage activity, similar to what would be observed for targeted dsDNA cleavage activity. One potential explanation for why this was not observed is that the dsDNA targets are not fully double stranded, and some ssDNA contamination is sufficient to activate LbCas12a. Such an issue would not arise for Cascade, given that ssDNA does not activate Cascade-Cas3 trans cleavage as well as a dsDNA target or a target containing a double-stranded PAM.

According to the suggestion, we added a following sentence in line 19-21, page 8, "One potential explanation for why the PAM specificity was not observed is that the dsDNA targets are not fully double stranded, and some ssDNA contamination is sufficient to activate LbCas12a."

Reviewer #2:

I support publications of this version. The authors have done a very good job of answering my questions and the community will appreciate the detail.

I do think the editor should ask for a bit more disclosure about the relationship between the company and this work, so that readers can make their own judgements about any potential bias.

According to this comment, we added disclosure about the relationship between the company and this work on page 33-34. Also, we have fully disclosed these interests to Nature Portfolio journals.

Reviewer #3:

The authors' appropriate revisions have satisfied all my questions, especially about terminology, the results of biochemical analysis, and their significance. The main HS-AFM image itself has not been updated, and its quality is still not entirely satisfactory, but the addition of the illustration helped me understand what the author was trying to state from the AFM images. I also commend the authors for the great effort they put into the quantitative analysis of the AFM data, which makes the HS-AFM data more convincing. On the other hand, what most of the results of the analysis of the HS-AFM data are included in the extended material raises me questions about the organization of the paper considering the description quantity in the main text. If it is possible due to the character limit of the journal, it would be better to move some HS-AFM data and analysis to the main text.

We appreciate these comments regarding to the HS-AFM images. According to them, we moved some HS-AFM data and analysis to the main text in a new Figure 5 in the revised manuscript.